

# 1  What do we know about how the terrestrial
# 2  multicellular soil fauna reacts to microplastic?

Frederick Büks[1], Nicolette Loes van Schaik[2], Martin Kaupenjohann[1]
[1]Chair of Soil Science, Dept. of Ecology, Technische Universität Berlin, 10587 Berlin, Germany
[2]Chair of Ecohydrology & Landscape Evaluation, Dept. of Ecology, Technische Universität Berlin, 10587 Berlin, Germany
*Correspondence to:* Frederick Büks (frederick.bueks@tu-berlin.de)
**Abstract.** The ubiquitous accumulation of microplastic (MP) particles across all global
ecosystems comes along with the uptake into soil food webs. In this review, we analyzed
studies on passive translocation, active ingestion, bioaccumulation and adverse effects within
the phylogenetic tree of multicellular soil faunal life. The representativity of these studies for
natural soil ecosystems was assessed using data on the type of plastic, shape, composition,
concentration and time of exposure.
Available studies cover a wide range of soil organisms, with emphasis on earthworms,
nematodes, springtails, beetles and lugworms, each focused on well known model organisms.
Most of the studies applied MP concentrations similar to amounts in slightly to very heavily
polluted soils. In many cases, however, polystyrene microspheres have been used, a
combination of plastic type and shape, that is easily available, but does not represent the
main plastic input into soil ecosystems. In turn, MP fibres are strongly underrepresented
compared to their high abundance within contaminated soils. Further properties of plastic
such as aging, coating and additives were insufficiently documented. Despite these
limitations, there is a recurring pattern of active intake followed by a population shift within the
gut microbiome and adverse effects on motility, growth, metabolism, reproduction and
mortality in various combinations, especially at high concentrations and small particle sizes.
For the improvement of future studies, we identified problems of past experiments and give
recommendations that take into account the type, shape, grade of aging, specific
concentrations of MP fractions and long-term incubation in natural and contaminated soils.



## 1 Introduction

Imagine a compact plastic cube of nearly 2 km side length and a weight of 7300000000 tons,
with major percentages by weight of 36 % polyethylene (PE), 21 % polypropylene (PP), 12 %
polyvinyl chloride (PVC) and 10 % of each polyethylene terephthalate (PET), polyurethane
(PU) and polystyrene (PS). That is the cumulated global non-fibre production of the six main
plastic types until 2015. It accounts to 87 % of the all-time plastic production, which evolved,
since the early 1950s, exponentially from some megatons (Mt) to 8300 Mt in 2015, with only
260 Mt annual output in 2009 increased to 380 Mt in 2015 (Thompson et al., 2009; Geyer et
al., 2017). Of this ever produced plastic, 6300 Mt became waste until 2015, of which only
21 % were recycled or incinerated, whereas 5000 Mt ended up in landfills and nature (Geyer
et al., 2017). As a corollary of production, use and disposal, a certain part of plastic waste is
constantly released into the environment on various paths, but our knowledge about mass
flow rates into global ecosystems is very limited. Based on waste generation in coastal
countries, Jambeck et al. (2015) calculated the global plastic input to marine ecosystems to
be roughly 4.8 to 12.7 Mt in 2010. Such data on soils are lacking, but Nizzetto et al. (2016)
estimated that the load of microplastic (MP) to agricultural sites in Europe is in the same order
of magnitude as to marine environments.
By littering, plastic mulching, the application of sewage sludge, digestates and composts as
well as windblown dispersal (Bertling et al., 2018; Weithmann et al., 2018; Zhang et al., 2019;
Wang et al., 2019a), plastic from our technosphere arrives in soil ecosystems in various forms
as large and small fragments, fibers and particles. Exposed to UV radiation, mechanical
stress and microbial decay, plastic items become weathered and prone to a successive
comminution towards the size range of MP with increased surface, charge and biofilm cover
(Kale et al., 2015; Andrady, 2017). However, the resistance of plastic to metabolization
causes a constant accumulation in soils as long as the release rate from human processes is
above the very slow rate of degradation.
Due to a lack of monitoring programs, data on MP concentrations in terrestric soils are rare,
and those using w/w concentrations are even sparser. Under less contaminated conditions,
amounts seem to average about 1 mg kg$^{-1}$ soil dry weight (and approx. 200 items kg$^{-1}$ dry soil)
(Rezaei et al., 2019). On sites with industrial activity or use of plastic mulching and sewage
sludge in agriculture, concentrations can be increased by 2 to 4 orders of magnitude (Fuller
and Gautam, 2016; Zhang and Liu, 2018). Semisubhydric soils such as beaches, mudflats,
mangroves or lagoons, that are additionally contaminated from the aquatic side, contain MP
of the order of 10 to 100 items kg$^{-1}$ dry soil and single extreme samplings contained several
thousand items (Nor and Obbard, 2014; Naji et al., 2017; Garcés-Ordóñez et al., 2019; Li et
al., 2018a). More informative data using mg kg$^{-1}$ are only available for beaches and coastal
deconstruction yards in municipal neighbourhood and amount to 0.5 and 70 mg kg$^{-1}$ dry soil,
0.00005 and 0.007 % w/w, respectively (Reddy et al., 2006; Claessens et al., 2011). All these



concentration data represent a wide range of particle sizes between 0 and 5000 µm with
different materials, shapes and grades of aging.
Plastic particles can possibly enter and accumulate within the food web by either direct
uptake from soil or consumption of other soil biota contaminated by adhesion or ingestion
(Huerta Lwanga et al., 2017a). There is evidence, that MP is incorporated even by plants and
unicellular organisms at the base of the food web. **Bacteria**, for example, that are reasonably
assumed to avoid MP uptake due to their minor size and the prevalent lack of phagocytosis,
were shown to take up inorganic nanoparticles of a few nanometers (Kumar et al., 2011).
Although the physiochemical properties of weathered nanoparticular plastics might differ from
these, also their uptake seems reasonable.
A similar argument can be made for **fungi** and soil **algae**, but studies on incorporation are
lacking, whereas the transfer into a freshwater food web by adhesion of nanoplastic on algae
has been shown by Chae et al. (2018). The uptake of MP into **plant roots** is also inhibited
(Rillig et al., 2019), but occured for nanoplastics that permeate into the plant tissue (Li et al.,
2019). Also the ingrowth into root tissue after adsorption to the rhizodermis should be tested.
In contrast, **protozoa** feature phagocytosis for the active ingestion of particles. Diverse soil,
freshwater and marine ciliates ingest PS/latex beads of 0.1 to 14.4 µm in laboratory
experiments, with preferences to their natural prey size (Fenchel, 1980; Jonsson, 1986; Lavin
et al., 1990). Soil amoebas act similarly, but additionally select according to food quality
(Weisman and Korn, 1967; Vogel et al., 1980; Bowers and Olszewski, 1983; Avery et al.,
1995; Elloway et al., 2006).
At last, many soil microbiota live protected within biofilms. Plastic particles were shown to be
surface for the formation of those biofilms (Lobelle and Cunliffe, 2011), which are a food
sources of grazing primary consumers. Feeding on them might also transfer occluded or
abrased MP to higher trophic levels.
But what about the larger organisms that feed on all these, free plastic particles,
contaminated microorganisms, biofilms and one another? Recent work discussed the effects
of MP on soil biota (Chae and An, 2018) or called for intensified research on certain
taxonomic groups (Rillig and Bonkowski, 2018). Thus, we were motivated to give on our part
a short review with focus on the most-produced plastics and their passive translocation,
ingestion, bioaccumulation and adverse effects on the multicellular soil fauna. The types,
sizes and shapes of plastic used in former laboratory studies were compared with our
knowledge on plastic in the environment, and recommendations are given for future research.
This analysis is aimed to help for assessing the influence of MP on the ecosystem services of
diverse soil organisms.



## 2 Search pattern

Within the tree of life, edaphic branches were identified comprising taxa that permanently inhabit the soil, are both-sided part of the soil food web and/or the burrowing macro- and megafauna or have active subterranean larval stages. The resulting tree of soil life based on the NCBI taxonomy database (Fig. 1) was charted by use of the software phyloT and shows the leading taxonomic rank, which is mainly the family, but in exceptions – e.g. if one species represents the only soil-born between many aquatic – a lower rank.

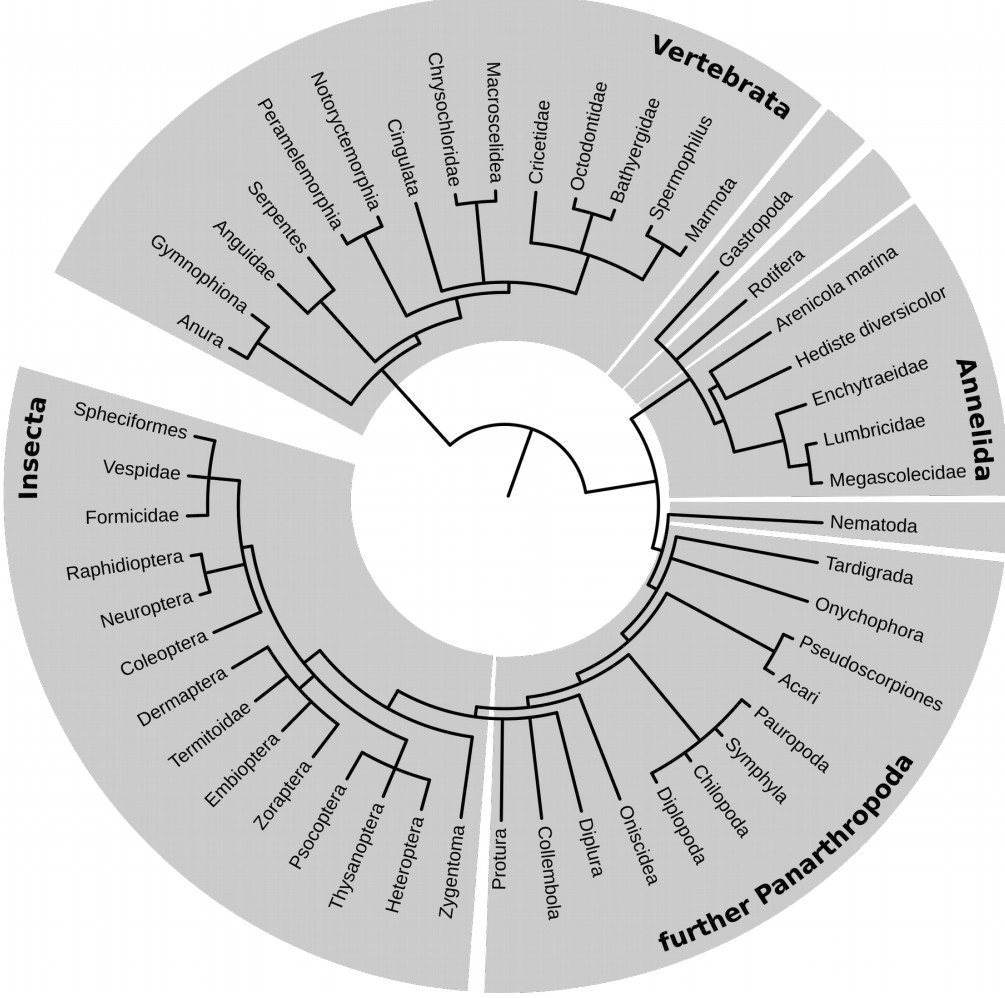

**Figure 1:** Edaphic tree of faunal life. Taxonomic ranks, that were examined in this qualitative study, are placed at the outer rim of the diagram. The length of the connecting line between two taxa is representative for the grade of phylogenetic relationship.



A pattern of search terms was established (see *Table 8*), consisting of „taxon" (Linné's
binominal nomenclature, common name, plural-sensitive search), „plastic type" (plastic,
microplastic, nanoplastic, PE or polyethylene, PP or polypropylene, PVC or polyvinyl chloride,
PS or polystyrene, PU or polyurethane, PET or polyethylene terephthalate and latex) and
„common shapes" (fragments, particles, fibres, microfibres, beads, microbeads,
microspheres). Type-shape combinations, that would had cause to much search effort (e.g.
organism–plastic) or did not appear within a foregoing search (e.g. PET–microbeads or latex–
microfibres), were excluded from this pattern. Further plastic types and shapes occuring
within the found studies were also included to the review. Data on microspheres and
microbeads were pooled, as both names describe one and the same.
The search appeared within the Web of Science Core Collection Database. Based on the
search pattern, data on passive transport, ingestion, bioaccumulation and adverse effects
were collected for each edaphic group. Studies that only use uncommon, local, outdated,
weird or nicknames are excluded by the pattern. Studies testing injection to tissues, lymph or
blood were excluded, as they do not represent natural ways to incorporate MPs. Data on
inhalation by the megafauna in fact represent a natural way of uptake, but were also excluded
as they are exclusively related to above-ground organisms, that only occur on the outer edge
of the food-web. Also running debates on phylogenetic classifications are not part of this work
and the taxonomist will be able to adjust the branches accordingly to his purpose.
The data of related taxonomic groups were pooled and evaluated for their environmental
representativity based on exposure time, plastic concentrations and properties used. From
these data recommendations for a structured experimental design in future studies were
derived.





## 3 Data collection

### 3.1 Insects

Within the Panarthropoda, the insects comprise the highest taxonomic diversity. And, regarding MPs, they represent an unevenly studied taxonomic group.

Within the Insecta, the **Coleoptera** (beetles) build an extraordinarily diverse and abundant taxon. Studies on plastic uptake into adult individuals mainly focus on the subfamily of Scarabaeinae (dung beetles). Comprehensive experiments with latex microbeads showed, that many species only ingest fine particles with maximum diameters of about 10 to 83 µm and retain them within the gut – with a slightly positive dependency on body size. Larger particles were rejected by a filtering mechanism within the mouth region and not ground with the mandibles (Holter, 2000; Holter et al., 2002; Holter and Scholtz, 2005). Beside those on Nematods, these data comprise by far the most detailed information about size-dependend uptake of MP particles compared to other edaphic taxa. This gives a good foundation for future studies on adverse concentrations. In addition, several studies with plastic as predominant food source could show chewing, ingestion and intestinal degradation of different PS and PE foams in feeding experiments with *Tenebrio sp.* larvae (mealworms). These experiments also pointed out an alteration of the gut microbiome, but no adverse effects on reproduction and survival, with only in one case of non-significant tendency to higher mortality after 1 month of exposure (Yang et al., 2015; Brandon et al., 2018; Yang et al., 2018; Peng et al., 2019).

The **Isoptera** (termites), recently categorized as part of the order Blattoidea, are the oldest social insects having a tribal history of about 130 million years (Korb, 2008). Especially in arid ecosystems with lack of earthworms they play an important role in homogenization of soils, but also in sorting of soil mineral particles for building mounds as well as decomposition and distribution of organic matter (De Bruyn and Conacher, 1990). Tsunoda et al. (2010) and Lenz et al. (2012) could show, that different termite species are picky feeders and erode PE, but avoid other plastic cable sheathings. This suggests the excretion of ground MP particles by termites, but metabolic impacts are unknown. In contrast to termites, data on **other Blattodea** (e.g. cockroaches) were not found.

The suborder **Apocrita** comprises some flying insects, that inhabit burrows within the soil, such as ground-dwelling wasps within the **Vespidae** superfamily, mining bees within the **Apoidea** superfamily and the **Spheciformes**. They mostly do not prey and feed on subterrestrial organisms, but may move MP particles into the ground, as implied by a report of Allasino et al. (2019) on soletary bees, which built nests fully made of plastic fragments. The Apocrita also contain the **Formicidae** (ants). Some ant species are considered an important factor for seed dispersal, a behavior, that could also be shown for artifical plastic seeds with ~2 mm diameter (Hughes and Westoby, 1992; Angotti et al., 2018). Robins and Robins (2011)



found that this also includes differently shaped cultural objects: *Rhytidoponera metallica*, a
representative of ground-nesting, omnivore ants, is capable not only of a remarkable
bioturbation but also of an active, apparently random burying of anthropogenic plastic
actefacts >1 mm. Seeds are used as a food source, thus, the ingestion of plastic bites is
conceivable, but not documented. The uptake of latex microspheres ≥0.88 µm with liquids by
larvae of *Solenopsis invicta* seems to be prevented by filtration within the mouth and the
particles are released as larger aggregates, whereas other species ingest by far larger
particles up to 150 µm (Glancey et al., 1981). However, also here data on adverse effects are
missing.
Further insects with edaphic adult stages, e.g. **Dermaptera** (earwigs), **Heteroptera** (true
bugs) and **Zygentoma** (silverfish, fishmoth, firebrat) or soil- or litter-dwelling larvae such as
**Embioptera** (webspinners, footspinners), **Thysanoptera** (thrips), **Psocoptera** (booklice,
barklice, barkflies), **Neuroptera** (lacewings)**, Raphidioptera** (snakeflies) or **Zoraptera** (angel
insects) are not yet researched with focus on soil MP.
Regarding insects, mainly studies on translocation and uptake of MP were carried out. In
contrast, work on bioaccumulation is completely lacking and adverse effects are sparsely
tested using *Tenebrio sp.* larvae. Such studies could provide information whether or not the
input of MP in soil ecosystems is one of many factors causing the global decline of the
entomofauna (Oliveira et al., 2019; Sánchez-Bayo and Wyckhuys, 2019).





**Table 1:** Microplastic studies on Coleoptera, Blattoidea (Blattoid.), Apoidea (A.) and Formicidae (mb=microbeads, fr=fragments, ms=microspheres, b=beads). Concentrations refer to mg kg$^{-1}$ dry soil, if not specially marked.

| | organism | experimental environment | plastic type | aging | coating | additives | shape | size span [µm] | concentrations | exposure time | passive transport | active uptake | bioaccum. dynamics | measured adverse effects | reference |
|---|---|---|---|---|---|---|---|---|---|---|---|---|---|---|---|
| Coleoptera | *Aphodius erraticus* | petri dish | latex | N/A | N/A | N/A | mb | 5 | N/A | 45 min | N/A | no | N/A | N/A | Holter (2000) |
| | *Aphodius rufipes* | | | | | | | 2.39 | | | | ≤14 µm | | | |
| | *Aphodius ater* | | | | | | | 2.39 | | | | ≤14 µm | | | |
| | *Aphodius fimetarius* | | | | | | | 2.39 | | | | ≤18 µm | | | |
| | *Aphodius contaminatus* | | | | | | | 2.39 | | | | ≤18 µm | | | |
| | *Aphodius fossor* | | | | | | | 2.39 | | | | ≤18 µm | | | |
| | diverse dung beetles | vial | latex | N/A | N/A | N/A | mb | 2.83 | N/A | 45 min | N/A | ≤10..≤60 µm | N/A | N/A | Holter et al. (2002) |
| | diverse dung beetles | N/A | latex | N/A | N/A | N/A | mb | 2.83 | N/A | 45 min | N/A | ≤4..≤95 µm | N/A | N/A | Holter and Scholtz (2005) |
| | *Tenebrio molitor* larvae | container | PS | N/A | N/A | no | foam | N/A | 100% w/w (food) | 31 d | N/A | yes | biodegrad. | N/A | Yang et al. (2015) |
| | *Tenebrio molitor* larvae | container | LD-PE PS | N/A | N/A | no flame retardant | foam | 8.27 cm³ | 50..100% w/w (food) | 32 d | N/A | yes | biodegrad. | microbiome | Brandon et al. (2018) |
| | *Tenebrio molitor* larvae | container | PS | N/A | N/A | N/A | foam | N/A | 4..100% w/w (food) | 32 d | N/A | yes | biodegrad. | no | Yang et al. (2018) |
| | *Tenebrio obscurus* larvae | N/A | PS | N/A | N/A | no | foam | N/A | 86..100% w/w (food) | 31 d | N/A | yes | biodegrad. | microbiome | Peng et al. (2019) |
| Blattoid. | *Coptotermes formosanus* | mesocosm | LD-PE others | yes/no | N/A | N/A | cable sheets | 4 cm, Ø 0.8 cm | N/A | 42 d | N/A | yes / no | N/A | N/A | Tsunoda et al. (2010) |
| | diverse termites | in situ | MD-PE PA | no | N/A | anti-oxidant stabilizer | cable sheets | 30 cm, Ø 1.4 cm | N/A | 6 yr. | N/A | yes / no | N/A | N/A | Lenz et al. (2012) |
| A. | *Megachile sp.* | in situ | N/A | N/A | N/A | N/A | fr | 0.9..4.5 | N/A | N/A | yes | N/A | N/A | N/A | Allasino et al. (2019) |
| Formicidae | *Solenopsis invicta* | petri dish | latex | N/A | N/A | fluorescence | ms | 0.9..4.5 | 2.5% w/w (food) | direct | N/A | filtration | N/A | N/A | Glancey et al. (1982) |
| | *Rhytidoponera metallica* *Aphaenogaster longiceps* *Pheidole sp.* | in situ | N/A | N/A | N/A | N/A | b | N/A | 50 items per nest | 3 d | yes | N/A | N/A | N/A | Hughes and Westoby (1992) |
| | *Rhytidoponera metallica* | mesocosm | N/A | N/A | N/A | N/A | diverse | <75.5 cm | N/A | 26 mos. | yes | N/A | N/A | N/A | Robins and Robins (2011) |
| | diverse ants | in situ | N/A | N/A | attractant | N/A | b | 1.8 cm | N/A | 1 d | yes | N/A | N/A | N/A | Angotti et al. (2018) |





## 3.2 Other panarthropods

Apart from the insects, **Acari** (mites) comprise many abundant soil-living taxa, that feed on litter, fungi and fauna as predators and parasites and are bioindicators, as they are sensitive to changes in the soil physiochemical environment (Gulvik, 2007). Experiments indicated, that mites passively transport MP due to pushing and dragging after attachment to their cuticle, as shown with 80 to 250 µm sized PVC particles in a petri dish experiment without soil (Zhu et al., 2018a). The population within manure pats slightly declines when exposed to mm-sized unweathered PE and PS particles at concentrations of 5 % v/v and declines strongly at ≥60 % v/v (Stamatiadis and Dindal, 1990). This could probably be an effect of moisture deficiency due to a reduced water holding capacity in an unnaturally enriched substrate, but not necessarily through plastic intake. In contrast, no data was found on their arachnoid, preying relatives, the order of **Pseudoscorpiones** (false scorpions).

Just as many other highly abundant and diverse representatives of the soil mesofauna, the **Oniscideae** (woodlice) contribute to the decomposition of litter by chewing and passage through their digestive system (Warburg, 1987) and react strongly to environmental pollution, as such they are potentially used as bioindicators (van Gestel et al., 2018). They practice a strict selection of natural food sources (Hassall and Rushton, 1984). This is also demonstrated for starch and cellulose based plastic films (4 cm²), which were consumed and digested in experiments with the model organism *Porcellio scaber*, in contrast to PHB (polyhydroxybutyrate) films, that reduces the feeding rate (Wood and Zimmer, 2014). Smaller PE particles (137±51 µm and 183±93 µm) embedded into food pellets (0.4 % w/w) were taken up easily by *Porcellio scaber*, and the smaller fraction caused a slight and non-significant reduction of body mass after 14 days of exposure, but not of feeding, defecation or energy reserves (Kokalj et al., 2018).

Other panarthropodean groups are even less studied in terms of MP. We did not find literature on the subphylum of Myriapoda containing the classes of **Diplopoda** (millipedes), **Chilopoda** (centipedes), **Pauropoda** and **Symphyla** (pseudocentipedes or symphilids), important litter-feeders and predators within various soil ecosystems.

The situation is nearly similar with the phylum of **Tardigrada** (water-bears or tardigrades), that has many ecologically relevant and well studied species feeding on microorganisms and detritus particles. Sparse field research in semisubhydric environments showed no uptake of MP fibres by tardigrada (Gusmão et al., 2016), but comprehensive data on terrestrial soils are lacking.

Another branch within the panarthropoda, the phylum of **Onychophora** (velvet worms), comprises primordial invertebrates that are mainly native in litter and soils with high water holding capacity under pleistocene-like forest vegetation within tropical and moderate regions (Monge-Nájera, 1994). As predators, they most likely take up plastic debris appearing within



or on their prey, but no studies on MP are available, most likely due to their remote habitats,
low abundance and little scientific focus.
The **Collembola** (springtails), an abundant, diverse and ubiquitous soil-borne phylum with a
broad spectrum of food sources (Hopkin, 1997), also represent an intensively studied group
within the Arthropoda. Together with the **Diplura** (which mainly live in tropic and subtropic
regions in litter and humid topsoil and feed on fungal hyphae, POM and prey) (Westheide and
Rieger, 1996) and the **Protura** (Pass et al., 2011), the Collembola build a morphological
group, that exhibit similar ecological functions, such as distribution and decomposition of
organic matter as well as the control of fungal abundance (Hopkin, 1997). Springtails provide
up to 27 % of the soil biomass and up to 33 % of the total soil respiration (with higher shares
in colder ecosystems) (Petersen, 1994) with up to 100000 individuals per square meter
(Hopkin, 1997). Thus, their well-being plays an important role for ecosystem functioning.
In a petri dish experiment without soil, Maaß et al. (2017) showed the passive transport of
urea-formaldehyde particles <400 µm and undefined PET fragments by two Collembola
species (*Folsomia candida* and *Proisotoma minuta*) due to attachment, but found no
ingestion. Within a soil matrix, trials of Kim and An (2019) indicated hindrance of collembolan
migration by larger PS particles (44±39, 282±131 and 676±479 µm) at concentrations of
1000 mg kg$^{-1}$ corresponding to highly contaminated soils. In addition, they found suppressed
mobility due to the attachment of even smaller PS microbeads (0.47 to 0.53 µm) at
concentrations of 8 mg kg$^{-1}$ dry soil, which is equivalent to values found in nature. Small
particles <50 µm were moved, while larger particles were most likely peeled off. When *F.*
*candida* encounters two of its predators, the mites *Damaeus exspinosus* and *Hypoaspis*
*aculeifer*, the dispersal of 80 to 250 µm PVC particles is enhanced as shown by Zhu et al.
(2018a) in a Petri dish experiment. Without proving the ingestion or the minimal effective MP
concentration, Zhu et al. (2018b) published an alteration of the gut microbiome and adverse
effects on growth and reproduction of *F. candida* by 80 to 250 µm PVC particles mixed in soil
at concentrations of 1000 mg kg$^{-1}$ dry soil. These data were not considered robust (van Gestel
and Selonen, 2018), but fit into a later study that found inhibited reproduction at
≥1000 mg kg$^{-1}$ and avoidance behavior as well as altered microbiome at ≥5000 mg kg$^{-1}$ (Ju et
al., 2019). Such concentrations can occur in highly contaminated soils (Fuller and Gautam,
2016). However, documentations on the active uptake, gnawing and grinding of MP by
collembolans proposed by Rillig (2012) is still lacking and also studies on Diplura and Protura.



**Table 2:** Microplastic studies on Acari, Oniscidea (Onisc.), Tardigrada (T.) and Collembola (fr=fragments, p=particles, mf=microfibres, mb=microbeads, ms=microspheres, [s]=semisubhydric). Concentrations refer to mg kg$^{-1}$ dry soil, if not specially marked.

| | organism | experimental environment | plastic type | aging | coating | additives | shape | size span [μm] | concentrations | exposure time | passive transport | active uptake | bioaccum. dynamics | measured adverse effects | reference |
|---|---|---|---|---|---|---|---|---|---|---|---|---|---|---|---|
| **Acari** | diverse mites | microcosm | PE PS | no | N/A | N/A | fr | <4800 >2000 | 0..90& v/v (manure) | 16 d | N/A | N/A | N/A | ≥5% v/v: abundance ↓ | Stamatiadis and Dindal (1990) |
| | Hypoaspis aculeifer Damaeus exspinosus | petri dish | PVC | N/A | no | N/A | p | 80..250 | 5000 items per dish | N/A | yes | N/A | N/A | N/A | Zhu et al. (2018a) |
| **Onisc.** | Porcellio scaber | mesocosm | PHB | no | N/A | N/A | fr | 4 cm² | 1 item per cosm | 14 d | N/A | yes | N/A | feeding ↓ | Wood and Zimmer (2014) |
| | Porcellio scaber | petri dish | PE | N/A | N/A | N/A | fr | 183±93 137±51 | 0.4% w/w (food) | 14 d | N/A | yes | N/A | no | Kokalj et al. (2018) |
| **T.** | diverse tardigrades [s] | in situ | N/A | N/A | N/A | N/A | mf | N/A | N/A | N/A | N/A | no | N/A | N/A | Gusmão et al. (2016) |
| **Collembola** | Folsomia candida Proisotoma minuta | cup | UF, PET | N/A | no | N/A | p,fr | <400 | 2.5..5 mg per cup | N/A | yes | N/A | N/A | N/A | Maaß et al. (2017) |
| | Folsomia candida | petri dish | PVC | N/A | no | N/A | p | 80..250 | 5000 items per dish | N/A | yes | N/A | N/A | N/A | Zhu et al. (2018a) |
| | Folsomia candida | microcosm | PVC | N/A | no | N/A | p | 80..250 | 1000 | 56 d | N/A | N/A | N/A | microbiome, growth ↓, reproduction ↓ | Zhu et al. (2018b) |
| | Folsomia candida | microcosm | PE | N/A | no | N/A | mb | <500 | 0..10000 0..10000 0..5000 | 7 d 28 d 28 d | N/A | N/A | N/A | ≥5000: avoidance ≥1000: reproduction ↓ ≥5000: microbiome | Ju et al. (2019) |
| | | | PS | N/A | carboxyl | fluorescence | mb | 0.5 | 4..8 | | yes | | | | |
| | | | PE | no | N/A | fluorescence | ms | 27..32 | 1000 | | yes | | | | |
| | | | PE | no | N/A | fluorescence | ms | 250..300 | 1000 | | N/A | | | | |
| | Lobella sokamensis | soil sample | PS | no | N/A | no | fr | 44±39 | 1000 | ≤3 min | yes | N/A | N/A | avoidance, motility ↓ | Kim and An (2019) |
| | | | PS | N/A | N/A | no | fr | 282±131 | 1000 | | N/A | | | | |
| | | | PS | no | N/A | no | fr | 676±479 | 1000 | | N/A | | | | |





### 3.3 Annelida

Another large group of invertebrates beside the branch of panarthropoda comprises land-based Annelida. Within the Annelida, the **Lumbricidae** (earthworms) comprise a well-studied family (Darwin, 1881; Lavelle et al., 2006), represented in high abundance and diversity in many ecosystems all around the world (Phillips et al., 2019). Earthworms are often used as indicators for soil health (Fründ et al., 2011; Pulleman et al., 2012), as they are ecosystem engineers which through their burrowing activity influence various soil physical, chemical and biological processes (Jouquet et al., 2006; Lavelle et al., 2006).

By far the most of the studies on the influence of MP on earthworms are performed with PE and the species *Lumbricus terrestris* or *Eisenia fetida*, but there are also single studies with *Aporrectodea rosea* (Boots et al., 2019) and *Eisenia andrei* (Rodriguez-Seijo et al., 2017) and with the less common species *Metaphire californica* (Wang et al., 2019b). We found one field study of earthworms and MPs (Huerta Lwanga et al., 2017a) among many laboratory experiments with MPs mixed into soil volumes (concentrations ranging up to 20000 mg kg$^{-1}$ dry soil) or applied with litter on top of the soil surface (≤60% w/w). The particles sizes were usually <1 mm in diameter, but some were even up to 2x2 cm², and the duration of experiments was generally 14 to 28 days, few lasted up to 60 days.

The uptake of MPs of a broad size range by earthworms was shown in studies based on particles in earthworm casts of *Lumbricus terrestris* (Huerta Lwanga et al., 2016; Cao et al., 2017; Hodson et al., 2017; Rillig et al., 2017; Prendergast-Miller et al., 2019; Yu et al., 2019; Huerta Lwanga et al., 2017a), *Eisenia fetida* (Rodríguez-Seijo et al., 2018; Chen et al., 2020; Wang et al., 2019c), *Eisenia andrei* (Rodriguez-Seijo et al., 2017) and *Metaphire californica* (Wang et al., 2019b). Zhang et al. (2018) showed that relatively large PE particles of 1.5 x1.5 cm² are not ingested by *Lumbricus terrestris*, but partial ingestion of such large particles of biodegradable MPs does take place after initial weathering in soil or in compost has occurred. In some laboratory experiments, MPs were found in the gut of dissected earthworms (Huerta Lwanga et al., 2016; Hodson et al., 2017; Rodriguez-Seijo et al., 2017), but the concentration of MPs in the gut was not significantly different between treatments nor significantly different from the bulk soil concentration, so there was no evidence of accumulation of MPs in the earthworm bodies (Hodson et al., 2017). Chen et al. (2020) assume an accumulation of MP takes place in *Eisenia fetida*, based on an observed increase of MP concentrations in the casts in the course of 4 weeks. Huerta Lwanga et al. (2017a) supposed an accumulation of MPs in the food chain as the concentration of MPs in chicken gizzards is strongly increased compared to that in the earthworm casts in the same experiments. However, mainly the amount of large particles, i.e. macroplastics, in the gizzards was very large, thus it seems likely that the chicken directly fed on plastics and an accumulation through the food chain cannot be proven with the current knowledge and should be further investigated.





Several studies did not find significant negative effects of MPs on earthworms' avoidance
behaviour (Judy et al., 2019), nor on growth (Hodson et al., 2017; Rodriguez-Seijo et al.,
2017; Judy et al., 2019; Wang et al., 2019c), mortality Hodson et al. (2017); Rillig et al.
(2017); Rodriguez-Seijo et al. (2017); Judy et al. (2019); Prendergast-Miller et al. (2019) or
reproduction (Huerta Lwanga et al., 2016; Rodriguez-Seijo et al., 2017). However, other
studies do show adverse effects of the uptake of MP in different degrees and on different
aspects of earthworms' fitness: A reduced growth was shown by Cao et al. (2017) for *Eisenia*
*Fetida* and the mortality increased at an exposure of concentrations ≥10000 mg kg$^{-1}$ dry soil.
At lower concentrations no significant effects were found. The growth of *Aporrectodea rosea*
was also inhibited when exposed to biodegradable polylactic acid, conventional high-density
polyethylene (at 1000 mg kg$^{-1}$ dry soil), and MP clothing fibers (at 10 mg kg$^{-1}$ dry soil) (Boots
et al., 2019). Huerta Lwanga et al. (2016) showed a decrease in growth and increased
mortality at concentrations ≥28% w/w in litter and after 60 days, though after just 14 days no
mortality occurred in these experiments.
In some studies, additional effects such as histopathological changes or stress biomarkers
were measured. For *Eisenia fetida* Chen et al. (2020) observed skin damage at
1500 mg MP kg$^{-1}$ in soil, measured an increase in catalase activity and malondialdehyde
content at 1000 mg kg$^{-1}$ and at ≥1000 mg kg$^{-1}$ acetylcholine esterase was significantly
stimulated. Wang et al. (2019c) tested *Eisenia fetida* and found that MPs only increased the
catalase and peroxidase levels as well as the level of lipid peroxidation and decreased the
activity of superoxide dismutase and glutathione S-transferase at an exposure of
200000 mg kg$^{-1}$ dry soil for 14 days. No discernible influence was found at 100000 mg kg$^{-1}$.
However, Rodríguez-Seijo et al. (2018) also found for *Eisenia fetida* a significant positive
correlation of MP concentration with different biomarker responses: catalase, glutathione S-
transferase, lactate dehydrogenase and thiobarbituric acid reactive substances. In addition,
Rodriguez-Seijo et al. (2017) observed histological damage of the gut and occurrence of
inflammatory processes as well as an increase of stress response indicators associated with
MP exposure of *Eisenia andrei*. For *Lumbricus terrestris* Prendergast-Miller et al. (2019)
showed an increase in metallothionein expression at an exposure with ≥1000 mg kg$^{-1}$ dry soil
and a decrease in heat shock protein 70 at a concentration of ≥10000 mg kg$^{-1}$.
Due to the large differences in experimental conditions – e.g. size of the MPs, addition of MPs
to soil or to litter, duration of experiments, earthworm species – the current knowledge is not
sufficient to detect whether there is a threshold in MP size and concentration at which the MP
become harmful for earthworms and how this threshold differs for different earthworms
species and MP shapes. The results of Huerta Lwanga et al. (2016), who found no effects of
MPs on earthworms at 14 days, but significant influence on growth and mortality after
60 days, indicate the importance of longer measurements. This is consistent with Pelosi et al.

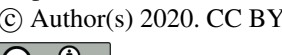


(2015), who concluded that the influence of pesticides on earthworm communities should be
tested in long term field experiments.
Earthworms activity also increased the transport of MP in soil columns to deeper soil layers
(Rillig et al., 2017; Yu et al., 2019; Huerta Lwanga et al., 2017b). The smaller the MP the
stronger the transport. Particles are transported both actively – ingested and later cast out –
and passively after attachment to the earthworm's body or by water flow through the biopores.
As Huerta Lwanga et al. (2018) showed that the bacteria in the gut of *Lumbricus terrestris* can
decompose MPs, it seems likely that particles taken up at the surface are egested as smaller
particles in deeper layers.
Microplastics might well serve as a vector for contaminant transport to soil organisms. Though
adsorption on plastics was seen to be lower than on the soil matrix, the desorption of Zn was
seen to be higher in synthetic earthworm guts. However, there was no measurable negative
effect of Zn or the PE on *Lumbricus terrestris* (Hodson et al., 2017). Wang et al. (2019b)
studied the influence of MP on arsenic uptake and negative effects on *Metaphire californica*
and concluded that MPs decreased the uptake of arsenic and that MPs reduced the influence
of arsenic on the gut bacterial communities. Rodríguez-Seijo et al. (2019) showed altered
enzyme activities and enhanced avoidance behavior in face of LD-PE pellets spiked with the
insecticide chlorpyriphos. Yang et al. (2019a) studied the influence of MPs on the transport of
glyphosate, however they mainly showed that the glyphosate transport was increased by
earthworm activity, the role of MPs in this transport could not be determined with this study.
These studies show that MP might have very different influences on the uptake and the
adverse effects of different pollutants on earthworms and further investigation is needed in
order to understand the influence of MPs on pollutant transport.
In contrast to the recently well-researched Lumbricidae, a near relative, the family of
**Megascolecidae** (giant earthworms), is not yet mentioned in literature. Another branch within
the Annelida, the small **Enchytraeidae** (potworms), were shown to suffer adverse effects on
body weight and microbiome with PS microspheres (0.05 to 0.1 µm) at concentrations of
≥10 % w/w within their food source, but an unexpected increase of reproduction at 0.5 % w/w
(Zhu et al., 2018b). The reproduction was reduced at abnormal concentrations of
90 g kg$^{-1}$ dry soil of polyamid particles (13 to 150 µm), but not with PVC (Lahive et al., 2019).
The edaphon of semisubhydric soils often became a marginal group between the area of
interest of soil and aquatic scientists. As a highly diverse soil biocoenosis outside the focus of
this paper, the benthos along seashores and fresh waters is also affected by MPs and should
therefore be shortly mentioned by reviewing the lugworm **Arenicola marina**, a well examined
deposit-feeder of the tidal flats. In situ, MP accumulates within its tissue and feces (Van
Cauwenberghe et al., 2015). In laboratory experiments, PS particles ≥500 µm were avoided
as food-source and passively translocated within the sediment at concentrations of ~2 g kg$^{-1}$





(Gebhardt and Forster, 2018), but were measured within the feces at ~74 g kg$^{-1}$ causing
effects on feeding activity and body weight with no influence on the survival rate (Besseling et
al., 2012). PS microspheres ≤30 µm remained within the animal without any adverse effects
regardless of particle size (Van Cauwenberghe et al., 2015). Other studies found adverse
effects on respiration, energy reserves, feeding, egestion and casting after uptake of PVC
particles ≤478 µm at different sediment concentrations of >2 g kg$^{-1}$, but neither due to HD-PE
nor on biomass and survival (Wright et al., 2013; Green et al., 2016). There is a difficulty in
distinguishing between the adverse effects of MPs and substances adsorbed on or leached
from MPs (Besseling et al., 2012). When adding PCB-spiked PE to mud flat sediment with
concentrations up to 5 g kg$^{-1}$ dry mass, there was no significant change of survival rate or
body weight. The decreased feeding activity and heap mass could be attributed to increasing
plastic concentrations, but not to enhanced PCB bioaccumulation via PE uptake (Besseling et
al., 2017). However, all these studies found adverse effects at MP concentrations orders of
magnitude above natural values.





**Table 3:** Microplastic studies on Lumbricidae (p=particles, ms=microspheres, b=beads, f=fibres, ms=microfibres). Concentrations refer to mg kg⁻¹ dry soil, if not specially marked.

| organism | experimental environment | plastic type | aging | coating | additives | shape | size span [µm] | concentrations | exposure time | passive transport | active uptake | bioaccum. dynamics | measured adverse effects | reference |
|---|---|---|---|---|---|---|---|---|---|---|---|---|---|---|
| Lumbricus terrestris | mesocosm | PE | washed ($C_6H_{12}$, $C_7H_{16}$) | N/A | N/A | p | <150 | 0..60% w/w (litter) | 14 d / 60 d | yes | yes | N/A | at 60 d, ≥28% w/w: survival↓, growth↓ | Huerta Lwanga et al. (2016) |
| Eisenia fetida | glass beaker | PS | N/A | N/A | N/A | ms | 50..80 | 0..20000 | 30 d | N/A | yes | N/A | ≥5000: survival↓ ≥10000: weight↓ | Cao et al. (2017) |
| Lumbricus terrestris | bag | PE | N/A | N/A | N/A | p | 0.92±1.09 mm² | 3500 | 28 d | N/A | yes | no | no | Hodson et al. (2017) |
| Lumbricus terrestris | home yard | diverse | yes | N/A | N/A | N/A | N/A | 0.87±1.9 items g⁻¹ | N/A | N/A | yes | conc. in chickens > in earthworms | N/A | Huerta Lwanga et al. (2017a) |
| Lumbricus terrestris | mesocosm | PE | washed ($C_6H_{12}$, $C_7H_{16}$) | N/A | N/A | p | <150 | 0..60% w/w (litter) | 14 d | yes | yes | N/A | N/A | Huerta Lwanga et al. (2017b) |
| Lumbricus terrestris | mesocosm | PE | N/A | no | no | b | 710..2800 | 750 µg on 2.5 kg soil | 21 d | yes | yes | N/A | no | Rillig et al. (2017) |
| Eisenia andrei | mesocosm | LD-PE | N/A | N/A | N/A | pellets | 250..1000 | 0..1000 | 28 d | N/A | yes | N/A | ≥62.5: intestinal damage | Rodriguez-Seijo et al (2017) |
| Lumbricus terrestris (gut bacteria) | mesocosm glass bottle | PE | washed ($C_6H_{12}$, $C_7H_{16}$) | N/A | N/A | p | 150 | 7% w/w (litter) 10000 | 60 d (earthworms) 21 d (bacteria) | N/A | yes | N/A | N/A | Huerta Lwanga et al. (2018) |
| Eisenia fetida | mesocosm | LD-PE | washed (EtOH) | N/A | N/A | pellets | 250..1000 | 0..1000 | 28 d | N/A | yes | N/A | ≥125: altered enzyme activity | Rodriguez-Seijo et al (2018) |
| Aporrectodea rosea | mesocosm | PLA, PE N/A | N/A | N/A | N/A | p f | N/A | 1.000 10 | 30 d | N/A | yes | N/A | growth↓ | Boots et al. (2019) |
| Eisenia fetida | mesocosm | HD-PE, PET, PVC | no | N/A | no | f | <2000 | soil extract | 48 h / 56 d | N/A | N/A | N/A | no | Judy et al. (2019) |
| Lumbricus terrestris | bag | PE | N/A | N/A | N/A | mf | ⌀40.7±3.8 x 361.6±387.0 | 0..10000 | 35 d | N/A | yes | N/A | ≥1000: metallothionein expression↑ ≥10000: heat shock protein 70↓ | Prendergast-Miller et al. (2019) |
| Eisenia fetida | mesocosm | LD-PE | washed (EtOH) | N/A | chlorpyrifos (CPF) | pellets | 5000 250..1000 | 40 items on 0.5 kg soil 180..200 items on 0.5 kg soil | 14 d | N/A | N/A | N/A | with CPF: altered enzyme activity, avoidance of MPs | Rodriguez-Seijo et al (2019) |
| Metaphire californica | mesocosm | PVC | N/A | N/A | sodium arsenate | p | N/A | 2000 | 28 d | yes | yes | N/A | microbiome | Wang et al. (2019b) |
| Eisenia fetida | glass beaker | PE PS | washed (MetOH) | N/A | PAHs, PCBs, Nile Red (NR) | p | <300 <250 | 0..200000 0..100 | 14 d 28 d | N/A | yes | N/A | ≥200000: altered enzyme activity | Wang et al. (2019c) |
| Lumbricus terrestris | mesocosm | PE | washed ($C_6H_{12}$, $C_7H_{16}$) | N/A | glyphosate | p | <150 | 0..7% w/w (litter) | 14 d | N/A | N/A | N/A | N/A | Yang et al. (2019a) |
| Lumbricus terrestris | mesocosm | PE | N/A | N/A | N/A | N/A | <1000 | 7% w/w (litter) | 14 d | yes | yes | N/A | N/A | Yu et al. (2019) |
| Lumbricus terrestris | petri dish mesocosm | PE and div. biodegradables | unweathered, field or compost | N/A | N/A | p | 1.5x1.5 cm² 2x2 cm² | 4 items per dish 10 items per dish | 14 d 50 d | yes | no yes | N/A | N/A | Zhang et al. (2018) |
| Eisenia fetida | bag | PE | washed (EtOH) | N/A | N/A | p | <400 | 0..1500 | 28 d | N/A | yes | yes | skin damage. ≥250 mg/kg: oxidative stress ≥1000 mg/kg: neurotoxicity↑ | Chen et al. (2020) |

Lumbricidae






**Table 4:** Microplastic studies on Enchytraeidae and *Arenicola marina* (mb=microbeads, p=particles, ms=microspheres, sed.=sediment, [s]=semisubhydric). Concentrations refer to mg kg$^{-1}$ dry soil in terrestrial and dry sediment in semisubhydric soils, if not specially marked.

| organism | experimental environment | plastic type | aging | coating | additives | shape | size span [µm] | concentrations | exposure time | passive transport | active uptake | bioaccum. dynamics | measured adverse effects | reference |
|---|---|---|---|---|---|---|---|---|---|---|---|---|---|---|
| *Enchytraeus crypticus* | petri dish | PS | N/A | N/A | N/A | mb | 0.05..0.1 | 0..10% w/w (food) | 7 | N/A | yes | N/A | at 0.5% w/w: reproduction ↑ ≥10% w/w: microbiome, weight ↓ | Zhu et al. (2018c) |
| *Enchytraeus crypticus* | microcosm | PA PVC | N/A | N/A | fluorescence N/A | p | 13..150 106..150 | 20000..120000 90000 | 20 h / 21 d | N/A | yes | N/A | ≥90000: reproduction ↓ no | Lahive et al. (2018) |
| *Arenicola marina*[s] | in situ | N/A | N/A | N/A | N/A | N/A | N/A | N/A | N/A | N/A | yes | 1.2±2.8 items g$^{-1}$ | N/A | Cauwenberghe et al. (2015) |
| *Arenicola marina*[s] | liquid culture | PS | no | N/A | N/A | ms | 10..90 | 10000..50000 items kg$^{-1}$ | 14 d | N/A | yes | 10 µm: 9600±1800 items kg$^{-1}$ 30 µm: 800±700 items kg$^{-1}$ | no | |
| *Arenicola marina*[s] | mesocosm | PS PA | yes | biofilm | N/A | p | 500..1000 | ~2000 ~1000 | 106..240 d | yes | no | N/A | N/A | Gebhardt and Forster (2018) |
| *Arenicola marina*[s] | mesocosm | PS | N/A | N/A | N/A | p | 400..1300 | 0..74000 | 28 d | N/A | ≥400 µm | no | ≥74000: feeding ↓, weight ↓ | Besseling et al., (2012) |
| *Arenicola marina*[s] | mesocosm | PVC HD-PE | N/A | N/A | N/A | p | 9..478 3..316 | 0..20000 mg kg$^{-1}$ wet sed. | 31 d | N/A | N/A | N/A | >2000: respiration ↓, casting ↓ no | Green et al. (2016) |
| *Arenicola marina*[s] | mesocosm | PE | N/A | PCBs | fluorescence | mb | 10..180 | 0..5000 | 28 d | N/A | yes | no | feeding activity ↓, heap mass ↓ | Besseling et al. (2017) |
| *Arenicola marina*[s] | mesocosm | PVC | N/A | N/A | not leaching | p | ~130 | 0..50000 | 28 d | N/A | N/A | N/A | ≥10000: energy reserves ↓ ≥50000: feeding ↓, egestion ↓, casting ↓ | Wright et al. (2013) |




### 3.4 Further invertebrates

As part of the microfauna, the phylum **Nematoda** (nematodes or roundworms) is an ecologically important branch containing >25000 species (Zhang, 2013) in freshwater, marine, endobiontic and soil habitats. Due to their diverse trophic interactions nematodes hold a central position in both bottom-up and top-down controlled food webs (Yeates, 2001; Ferris, 2010) and thus most likely the uptake and transfer of MP.

Active feeding of adults and larvae of different species on 0.5 to 6 µm PS/latex microspheres (the size of their bacterial prey) was proven by Nika et al. (2016) and Fueser et al. (2019). However, most MP experiments on Nematodes are based on the bacterial-feeding model organism *Caenorhabditis elegans*. Kiyama et al. (2012) showed the favored uptake of PS microspheres with sizes of 0.5 to 3 µm by adult and 0.5 µm by larval *C. elegans*. The ingestion of MP decreased in the presence of bacteria as the natural food source.

When larval stages and adults ingested PS between 0.05 and 5 µm within an aqueous suspension or on agar plates, adverse effects such as oxidative stress, neurodegeneration, intestinal and DNA damage or dysfunction in motility, growth, life span, defecation, reproduction or energy metabolism appeared from a wide spectrum of concentrations from ≥1 µg l$^{-1}$ up to ≥86.3 mg l$^{-1}$ (Zhao et al., 2017; Dong et al., 2018; Kim et al., 2019; Lei et al., 2018a; Lei et al., 2018b; Qu et al., 2019a). These effects are missed below 1 µg l$^{-1}$ (Qu et al., 2019b), and are enhanced due to amino modifications on microsphere surfaces (Qu et al., 2019c). The incubation on agar plates with PE, PP and PVC particles <70 µm caused similar influences on survival, fertility, brood size and intestinal function (Lei et al., 2018b) Leachates from soils amended with 5 mg kg$^{-1}$ dry soil of HD-PE and PVC decreased reproduction in laboratory cultures, but there was no effect shown on survival and after application of PET (Judy et al., 2019). Furthermore, silica nanoparticles (0.05 µm) are not only taken up orally but also via the vulva and spermathecae and migrate into gonad cells (Scharf et al., 2013), This process was confirmed for PS nanoparticles with the potential of a transfer to the progeny (Zhao et al., 2017).

The clear adverse effects of these studies are limited in their representativity by a narrow restriction to liquid cultures and a single model organism lacking broader studies on prominent soil-born nematodes such as *Acrobeloides buetschlii* (Frey, 1971). When assuming in first proximity mg l$^{-1}$ solution = mg kg$^{-1}$ dry soil, the applied concentrations between 0.001 and 86.8 mg l$^{-1}$ match lower levels of soil contamination.

Feeding studies on the phylum **Rotifera** with MPs are fully based on PS microbeads and model organisms of the planktonic genus *Brachionus*. However, this data can carefully be transferred to soil environments as also soil rotifers are aquatic organisms living in water-filled pores and waterfilms. Different *Brachionus sp.* ingest microbeads <10 µm with strong preference for particles the size of their natural food source, namely bacteria and algae with





2 to 5 µm in diameter (Vadstein et al., 1993; Heerkloß and Hlawa, 1995; Baer et al., 2008;
Jeong et al., 2016). The uptake appears to be selective as the microbeads are less
incorporated than bacteria and algae (Vadstein et al., 1993). The egestion of particles
≤0.5 µm is hindered compared to 6 µm (Jeong et al., 2016). In suspension, microbeads
≤0.5 µm cause adverse effects on fertility and life span at ≥0.1 mg l$^{-1}$ as well as oxidative
stress and less growth at ≥10 mg l$^{-1}$ (Jeong et al., 2016; Sun et al., 2019).
Terrestrial molluscs comprise snails and slugs within the class of **Gastropoda**. These grazers
feed on bacterial biofilms, fungi and plant tissue (Parkyn and Newell, 2013). Studies on
terrestrial species are sparse, but data on the benthic *Littorina sp.* imply passive transport and
non-selective MP uptake by feeding on surfaces with contaminated feces and mucus trails of
other snails (Gutow et al., 2019). However, Imhof and Laforsch (2016) found no significant
influence on growth parameters and fertility of juveniles and adult *Potampoyrgus antipodarum*
even when a food source with 70 % w/w of 5 to 600 µm sized fragments was given (a mixture
of PA, PC, PET, PS, PVC). In contrast, adverse effects were found in recent work on the
terrestrial snail *Achatina fulica*, that showed uptake and complete gastrointestinal passage
within 48 h with partial degradation of PET fibres (appr. 1258x76 µm), but reduced excretion
and food intake as well as increased oxidative stress at concentrations of ≥0.01 g kg$^{-1}$,
≥0.14 g kg$^{-1}$ and ≥0.71 g kg$^{-1}$ dry soil, respectively (Song et al., 2019).

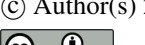



**Table 5:** Microplastic studies on nematods (ms=microspheres, fr=fragments, np=nanoparticles, mb=microbeads, ms=microspheres, ox.=oxidative). Concentrations refer to mg kg$^{-1}$ dry soil, if not specially marked.

| organism | experimental environment | plastic type | aging | coating | additives | shape | size span [µm] | concentrations | exposure time | passive transport | active uptake | bioaccum. dynamics | measured adverse effects | reference |
|---|---|---|---|---|---|---|---|---|---|---|---|---|---|---|
| *Caenorhabditis elegans* | agar plate | PS | N/A | carboxyl sulfate amino | fluorescence | ms | 0.1..6.6 | N/A | 0.5..2 h | N/A | yes | 0.5..3 µm | N/A | Kiyama et al. (2012) |
| *Caenorhabditis elegans* | liquid culture | PS | N/A | carboxyl | fluorescence | ms | 0.1 | 0.001..10 mg l$^{-1}$ | 4.5 d | N/A | Yes | N/A | ≥0.01 mg l$^{-1}$: motivity ↓, growth ↓, defecation ↓, within gonads | Zhao et al. (2017) |
| *Caenorhabditis elegans* | liquid culture | PS | N/A | ζ=-10mV | fluorescence | ms | 0.1 | 0.00001..0.001 mg l$^{-1}$ | N/A | N/A | Yes | N/A | ≥0.001 mg l$^{-1}$: motivity ↓, ox. stress ↑ | Dong et al. (2018) |
| *Caenorhabditis elegans* | liquid culture | PS | N/A | N/A | preservatives, fluorescence | ms | 0.05..0.2 | 0.001..86.8 mg l$^{-1}$ 17.3..86.8 mg l$^{-1}$ | 24 h | N/A | Yes | N/A | ≥17.3 mg l$^{-1}$: motivity ↓,fertility ↓ ≥86.3 mg l$^{-1}$: ox. stress ↑ ≥17.3 mg l$^{-1}$: metabolic dysf. | Kim et al. (2019) |
| *Caenorhabditis elegans* | liquid culture | PS | N/A | ζ=-10mV | fluorescence | ms | 0.1 | 0.001..1 mg l$^{-1}$ | N/A | N/A | Yes | N/A | ≥1 mg l$^{-1}$: neurodegeneration ≥0.01 mg l$^{-1}$: motivity ↓ | Qu et al. (2019a) |
| *Caenorhabditis elegans* | liquid culture | PS | N/A | N/A | N/A | ms | 0.1..5 | 1 mg l$^{-1}$ | 3 d | N/A | Yes | N/A | motivity ↓, survival ↓, growth ↓, ox. stress ↑, neurotoxicity | Lei et al. (2018a) |
| *Caenorhabditis elegans* | agar plate | PE, PP, PVC, PS | no | N/A | N/A | fr, ms | 0.1..200 | 0.5..10.0 mg m$^{-2}$ | 2 d | N/A | Yes | N/A | ≥0.5 mg m²: survival ↓ at 5 mg m²: growth ↓, fertility ↓, ox. stress ↑, intestinal damage mainly 1µm: intestinal damage | Lei et al. (2018b) |
| *Caenorhabditis elegans* | agar plate | PS | N/A | N/A | fluorescence | ms | 0.05 | | | | | | | |
| *Caenorhabditis elegans* | agar plate | silica gel | N/A | N/A | N/A | np | 0.05 | 2500 mg l$^{-1}$ | 7 d | N/A | Yes | N/A | within tissue and gonades | Scharf et al. (2013) |
| *Caenorhabditis elegans* | liquid culture | HD-PE, PET, PVC | no | N/A | no | fr | <2000 | soil extract | 72 h | N/A | N/A | N/A | fertility ↓ | Judy et al. (2019) |
| *Caenorhabditis elegans* *Panagrolaimus thienemanni* *Plectus acuminatus* | agar plates | latex | N/A | N/A | fluorescence | mb | 0.5 | N/A | 30 min | N/A | yes ≤3µm ≤0.5µm | N/A | N/A | Nika et al. (2016) |
| *Poikilolaimus regentussi* *Acrobeloides nanus* *Pristionchus pacificus* *Aphelenchoides parietinus* | liquid culture | PS | N/A | N/A | fluorescence | ms | 0.5..6 | 3·10⁹..10¹⁰ items l$^{-1}$ (~0.2..1200 mg l$^{-1}$) | 4..73 h | N/A | ≤1µm ≤1µm ≤6µm no | N/A | N/A | Fueser et al. (2019) |
| *Caenorhabditis elegans* | liquid culture | PS | N/A | N/A | N/A | ms | 0.1 | 0.0001..0.001 mg l$^{-1}$ | N/A | N/A | N/A | N/A | no | Qu et al. (2019b) |
| *Caenorhabditis elegans* | liquid culture | PS | N/A | no amino | N/A | ms | 0.1 | 0.001..1 mg l$^{-1}$ | N/A | N/A | yes | N/A | ≥0.01 mg l$^{-1}$: fertility ↓, DNA damage ≥0.001 mg l$^{-1}$: fertility ↓, DNA damage | Qu et al. (2019c) |

Nematoda






**Table 6:** Microplastic studies on Rotifera and Gastropoda (ms=microspheres, mb=microbeads, fr=fragments, f=fibres, ox.=oxidative, pref.=preferential, p=planctic, b=benthic). Concentrations refer to mg kg⁻¹ dry soil, if not specially marked.

| organism | experimental environment | plastic type | aging | coating | additives | shape | size span [µm] | concentrations | exposure time | passive transport | active uptake | bioaccum. dynamics | measured adverse effects | reference |
|---|---|---|---|---|---|---|---|---|---|---|---|---|---|---|
| **Rotifera** | | | | | | | | | | | | | | |
| Brachionus plicatilis [p] | liquid culture | PS | N/A | carboxyl | fluorescence | ms | 1.6..20 | $5 \cdot 10^9$ µm³ l⁻¹ ($\sim$5.25 mg l⁻¹) | 35 min | N/A | ≤10 µm | pref. 4.5 µm | N/A | Bear et al. (2008) |
| Brachionus plicatilis [p] | liquid culture | latex | N/A | N/A | fluorescence | mb | 0.3..3.1 | $3 \cdot 10^7 .. 7 \cdot 10^8$ items l⁻¹ ($\sim$0.0004..11 mg l⁻¹) | 20 min | N/A | yes | pref. ≥2 µm | N/A | Vadstein et al. (1993) |
| Brachionus koreanus [p] | liquid culture | PS | no | N/A | fluorescence | mb | 0.05..6 | 0....20 mg l⁻¹ | 1 d | N/A | yes | egestion rate 0.05 µm < 0.5 µm < 6 µm | ≤0.5 µm, ≥0.1 mg l⁻¹: fertility ↓, survival ↓; ≤0.5 µm, 10 mg l⁻¹: oxidative stress ↑ | Jeong et al. (2016) |
| Brachionus plicatilis [p] | liquid culture | PS | N/A | N/A | N/A | mb | 0.07..7 | 0..20 mg l⁻¹ | N/A | N/A | yes | N/A | ≤0.07 µm, ≥10 mg l⁻¹: fertility ↓, growth ↓; ≤0.07 µm and ≥0.1 mg l⁻¹: survival ↓ | Sun et al. (2019) |
| Brachionus quadridentatus [p] Brachionus plicatilis [p] | liquid culture | PS | N/A | N/A | N/A | ms | 2..10 | N/A | 8..10 d | N/A | pref. 3..5 µm pref. 2 µm | N/A | N/A | Heerkloß and Hlawa (1993) |
| **Gastropoda** | | | | | | | | | | | | | | |
| Littorina littorea [b] | microcosm | PMMA | N/A | N/A | fluorescence | fr | 10..100 | increasing | 16 h | N/A | yes | N/A | N/A | Gutow et al. (2019) |
| Potampoyrgus antipodarum [b] | aquarium | PET, PS, PVC, PA, PC | N/A | N/A | no | fr | 5..600 | 0..70% w/w (food) | ≤141 d | N/A | yes | N/A | no | Imhof and Laforsch (2016) |
| Achatina fulica | mesocosm | PET | N/A | N/A | no / stained | f | approx. 1258x76 µm | 10..710 | 28 d | N/A | yes | excretion after 48 hours | ≥140: food intake ↓; ≥10: excretion ↓; ≥710: ox. stress ↑, gastrointestinal damage | Song et al. (2019) |



## 3.5 Vertebrates

Different taxa of the class of Amphibia have a predator function within the edaphic food web (e.g. preying on invertebrates) (Hebrard et al., 1992). While no data on the reaction to soil MPs are available neither for the legless **Gymnophiona** nor for adults of the order **Anura**, sparse data on tadpoles of aquatic frogs suggest uptake followed by regular excretion of PS microspheres as shown with *Xenopus tropicalis* (Hu et al., 2016). Further, there exist no data on the families **Serpentes** (snakes) and **Anguidae** within the class of Reptilia, residing at the outer rim of the food web.

Within the broad field of Mammalia, studies on MP ingestion are sparse and focus on **mice** as a rodent model organism. Feeding of mice with PS microspheres of 1 to 14 µm in concentrations of $1.49x10^6$ to $4.55x10^7$ particles at a volume of 10 ml $kg^{-1}$ body weight for 4 weeks showed no adverse effects (Stock et al., 2019). In contrast, longer exposition (6 weeks) with lower concentrations of particles with the same shape and size range changed the mouse microbiome and caused metabolic and intestinal dysfunction (Lu et al., 2018; Jin et al., 2019), which comes along with bioaccumulation within organs (Yang et al., 2019b). These studies are regularly conducted with passive feeding and exclude active foraging on perceptible plastic particles. However, the uptake via prey or feeding on contaminated roots and litter is highly probable. Further Rodentia – **Cricetidae** (hamsters, lemmings, voles)**, Bathyergidae** (blesmols, mole-rats), **Octodontidae** as well as **Spermophilus** (ground squirrels) and **Marmota** (marmots) within the family of **Sciuridae** – were not yet studied, just as other mammalian (sub)orders like **Chrysochloridae** (golden moles), **Cingulata** (armadillos), **Macroscelidea** (elephant shrews), **Notoryctemorphia** and **Peramelemorphia**.





**Table 7:** Microplastic studies on Anura (An.) and Rodentia (ms=microspheres, $^a$=aquatic).

| organism | experimental environment | plastic type | aging | coating | additives | shape | size span [µm] | concentrations | exposure time | passive transport | active uptake | bioaccum. dynamics | measured adverse effects | reference |
|---|---|---|---|---|---|---|---|---|---|---|---|---|---|---|
| **An.** *Xenopus tropicalis* $^a$ | petri dish | PS | N/A | N/A | fluorescence | ms | 1..10 | $100..10^8$ items $l^{-1}$ ($55·10^9..55$ mg $l^{-1}$) | 48 h | N/A | yes | egestion within days | N/A | Hu et al. (2016) |
| **Rodentia** transgenic mice | in vivo | PS | N/A | carboxyl | fluorescence | ms | 1 | $4.55·10^7$ items per mouse (0.025 mg per mouse) | 28 d | N/A | yes | N/A | no | Stock et al. (2019) |
| | | | | sulfate | | | 4 | $4.55·10^7$ items per mouse (1.6 mg per mouse) | | | | | | |
| | | | | sulfate | | | 10 | $1.49·10^6$ items per mouse (0.8 mg per mouse) | | | | | | |
| *mice* | in vivo | PS | N/A | N/A | fluorescence | ms | 5 | $0.1..1$ mg $l^{-1}$ | 42 d | N/A | yes | N/A | $\geq0.1$ mg $l^{-1}$: microbiome, metabolic dysfunction | Jin et al. (2019) |
| *mice* | in vivo | PS | N/A | N/A | N/A | ms | 0.5..50 | $0.1..1$ mg $l^{-1}$ | 35 d | N/A | N/A | N/A | $\geq0.1$ mg $l^{-1}$: microbiome, metabolic dysfunction; $\geq1$ mg $l^{-1}$: body weight ↓ | Lu et al. (2018) |
| *Mus musculus* | in vivo | PS | N/A | N/A | fluorescence | ms | 5..20 | 200 mg $l^{-1}$ | 28 d | N/A | yes | 8x, 8±5 and 0.71±0.14 mg kg$^{-1}$ body weight | N/A | Yang et al. (2019b) |





## 4 Synthesis

### 4.1 Summarized observations

Our systematic search comprised recent research on the interaction of soil organisms with MP, but also studies with focus on feeding experiments, that are published much earlier than the awareness on plastic in the environment appeared. The numerous studies found with focus on the ingestion of MPs consistently showed the active uptake by diverse soil organisms with few exceptions spread over the whole branch of invertebrates. In addition, also studies on adverse effects caused by the intake of MP contaminated food (e.g. of food pallets by dung beetles) imply the ingestion into the test organism. Distinct size preferences are measured for dung beetles, nematodes, rotifers and ants showing that mainly particles are ingested, that are small enough to enter the gastrointestinal tract. In contrast, active comminution by gnawing on larger particles was tested only for a few taxa and confirmed for woodlice, termites and mealworms, and in the case of earthworms only after initial weathering.

After the ingestion, MP is understandably translocated actively until excretion or death of the transporting organism, which was only directly shown in experiments with earthworms. The passive transport by attachment, dragging and pushing was checked in a few experiments with earthworms, mites and springtails that partly worked without soil substrate and consistently showed positive results.

After exposition to MP, a pattern of adverse effects can be seen: Across various taxa, altered microbiomes, reduced motility, body mass, fertility and life span as well as increased oxidative stress and metabolic malfunctioning occur in different combinations mainly due to μm-sized MP in and above the whole known natural range of concentration. For some taxa such as Nematodes, Gastropoda and Rotifera these effects appear at natural and increased MP concentrations (<100 mg kg$^{-1}$ dry soil), for Collembola and Lumbricidae at concentrations like in highly contaminated sites (≥1000 mg kg$^{-1}$ dry soil) and for Enchytraeidae, *Arenicola marina* and in further experiments with earthworms at unplausible high values. The data show a tendency, that the effects occur at lower concentrations, when the added particles are smaller. Small sized particles also provide the highest surface/volume ratio and thus the highest reactive surface per weight.

Most studies work with defined increasing MP concentrations and particle sizes in soil substrates and food sources, which can be used to determine relationships between environmental concentrations and adverse effects. However, the lack of information about intake rates, grades of accumulation and effective prey-predator transfer leads to a gap within the chain of explanation for toxic effects on the soil organisms. In some experiments, the intestinal passage of MP and sizes preferably retained within the gut were shown, but there are no experiments that could demonstrate quantitative bioaccumulation. In contrast,



quantifications of the retained and egested MP particle size fractions might be biased due to
gnawing and intestinal comminution as shown for woodlice, termites, mealworms, snails and
earthworms.
In order to improve our understanding of processes underlying adverse effects of MP on soil
organisms, data on ingestion rates, dwell times, biodegradation and egestion rates are
important bricks e.g. to reveal bioaccumulation dynamics. However, there are only a few data
on biodegradation (mealworms, snails, earthworms), egestion (rotifers, frogs, snails,
earthworms) and remaining concentrations in the body (lugworm, mice, earthworms).





**4.2 Limitations of previous studies**

The available studies worked with items within the full size span of micro- and nanoplastics (≤5000 μm). When MP ≥50 μm was applied, mainly particles and fragments made of PE and PVC were used, whereas PS/latex microspheres were mainly applied for sizes ≤10 μm (*Table 8*). The latter are readily available, highly standardized and are mostly used with fluorescent dyes and either hydrophobic, carboxylated or, more rarely, with amino or sulfate groups. However, there are indications that the spectrum of particle type and shape used in experiments does not correspond to the properties of particles in soils. In different natural as well as agriculturally and industrially contaminated terrestrial and semi-subhydric sites, fibers and fragments of PE and PP, mostly ≤100 μm, were much more abundant than PVC, PET and PS items (Claessens et al., 2011; Vianello et al., 2013; Nor and Obbard, 2014; Naji et al., 2017; Zhang and Liu, 2018; Li et al., 2018a). This is probably caused by high loads of MP fibers in discharged waste water and sewage sludge, which is used in agricultural sites worldwide (Mahon et al., 2016; Li et al., 2018b). It is likely that shape plays an important role for the ingestion of MP items. Unfortunately, we did not find studies that have carried out a complete classification of sampling sites according to plastic origin, size and type, that could help to evaluate differences between former experimental and natural plastic composition to achieve the most realistic experimental conditions. Little knowledge about the size distribution of MP in soils furthermore complicates the determination of realistic concentrations for the addition of a certain particle size spectrum. All reviewed studies either arbitrarily set their applied concentrations or had to base them on measurements of total specific MP masses, regardless of how much of this mass is in the tested size range. This may lead to a malestimation of total adverse MP concentrations.

In contrast to particle type and shape, the documentation of chemical properties of MP samples in most of these studies is fragmentary. Some experiments explicitly mentioned that the added plastic was unweathered, whereas most studies lack of information about the degree of aging implying that unweathered items were used. Only a few experiments involved aging of MP, but without comparison to results of natural weathering (Tsunoda et al., 2010; Gebhardt and Forster, 2018). That is in conflict with natural conditions, as plastic that remaines within the soil after littering, sewage sludge application or plastic mulching shows signs of weathering, e.g. modified carbonyl indices (Andrady, 2017), while unweathered soil MP might be rare. In addition, Zhang et al. (2018) showed that earthworms actively comminute only weathered bioplastics. In experiments using PS microspheres, the reduced hydrophobicity due to weathering is therefor imitated by means of surface carboxylation.

Weathering of MP surfaces within soils comes along with biofilm growth and adsorption of organic molecules, which could potentially affect the attractiveness or toxicity for grazers and other organisms. Such coatings were applied only in a few cases (Besseling et al., 2017; Angotti et al., 2018; Gebhardt and Forster, 2018), but were not documented in most studies.





Similarly, the type and concentration of additives such as flame retardants, anti-oxidants or
stabilizers often remained undocumented, with exception of fluorescent dyes, that are well
mentioned. The release of additives can have a harmful effect on the test organism, as shown
for aquatic environments (e Silva et al., 2016). Some studies on the ingestion of MP by the
soil mesofauna indicate that the diameter of the gastrointestinal tract is a useful upper size
limit for added particles, as far as the organism is unable to crush them (Heerkloß and Hlawa,
1995; Holter, 2000; Holter et al., 2002; Holter and Scholtz, 2005; Baer et al., 2008; Fueser et
al., 2019). However, using only ingestible particle sizes in their natural concentrations neglect
the adverse effects of plastic leachates, which can also get into the soil solution and onto the
mineral phase from larger particles and affect soil life.
The conditions of incubation differ considerably in terms of habitats and duration of exposure.
In most studies, the exposure ranges from a few minutes to a few days in experiments with
micro- and small mesofauna and hours to several weeks in experiments with large meso- and
macrofauna and is mainly based on excretion or reproductiv cycles. Long-term studies, which
are indeed difficult to carry out in mesocosms, practically do not exist. However, certain
adverse effects might only establish themselves after long term trials, as was shown for the
influence of pesiticides (Pelosi et al., 2015).
Some experiments were carried out in soil-free test environments such as liquid cultures or
petri dishes with nutrient solutions or a specific food source (nematods, rotifers, mice). By
this, motivity is less restricted and feeding behavior can be altered compared to cultivation
within soil environments. For example, the ingestion of MP by nematods decreases in the
presence of an alternative and more natural food source like bacteria, which can significantly
reduce the bioaccumulation and thus the effective toxicity (Kiyama et al., 2012). This can lead
to less consumption of MP in soil environments and an overestimation of the toxicity in liquid
culture experiments. Also, all laboratory feeding experiments were carried out by use of only
one species. The complexity of the food web in soils is thereby excluded and the potential
accumulation from prey to predators still unexplored.






**Table 8:** Types and shapes of microplastic particles in edaphon studies. (X) symbolizes combinations outside the search pattern, the number counts for studies, empty fields stand for zero results. Microbeads and microspheres are often mixed up terms and, thus, counted together.

| Linné's systematic names OR common name | fragments | particles | fibres | microfibres | beads microbeads | microspheres | other, diverse, N/A |
|---|---|---|---|---|---|---|---|
| plastic | X | | | | | | |
| microplastic | | | | | | | |
| nanoplastic | | | | | | | |
| PE *OR* polyethylene | X | 4 | 10 | 1 | 1 1 | 4 | 7 |
| PP *OR* polypropylene | X | 1 | | | | | |
| PVC *OR* polyvinyl chloride | X | 4 | 6 | 1 | | | |
| PS *OR* polystyrene | X | 6 | 3 | | | 24 | 4 |
| PU *OR* polyurethane | X | | | | | | |
| PET *OR* polyethylene terephthalate | X | 3 | | 2 | | X | |
| latex | X | | | | X | 6 | |
| other | | 6 | 3 | 1 | | | 1 |
| N/A | | 1 | | 1 | 2 | | 3 |






### 4.3 Pinpoints for future research

Most studies reviewed in this work have a pioneering role in MP research and, thus, are subject to some experimental limitations caused by an early state of knowledge. The adverse effects recently found are alarming, but must be considered under the restrictions named above. We propose the following points as part of a *modus operandi* for future MP research.

In past studies, particular adverse effects of MP were measured only for certain sizes, shapes, coatings, leachates or adsorbed substances (*Tables 1 to 7*). Experimental concentrations were assumed randomly or derived from cumulative concentrations of one or more MP types measured in natural soils (approx. 1 to some 1000 mg kg$^{-1}$ dry soil), regardless of size. For those specific experiments coming, the spectrum of concentrations used should be adapted to the quantities of the size spectrum occuring within the soil. For future studies on mixed contaminations, we recommend to evaluate the overall adverse effects of PE, PP, PVC, PET, PU and PS to certain test organisms by use of typical MP-specific spans of concentration, size and shape distribution in natural soils or food samples. This previously requires well-structured data of appropriate MP type, shape and size for different soils in differently contaminated areas.

Experiments on adverse effects should be applied within soil matrices to allow the interplay of plastic, natural organic and mineral matter. The MP should be weathered, as plastic in soils underlie broad environmental aging. Pre-weathering of MP should therefor not only be performed in climate chambers (e.g. following DIN EN ISO 4892-2/3), but also include subsequent leaching and equilibration of additives or coatings within the soil matrix before the main experiment. Furthermore, the experimental design may consider coatings with biofilms or attractants and even particle color to regulate the preference of the test organisms.

Most detailed information about ingestion are available for dung beetles, nematods and earthworms, data on adverse effects on nematods, earthworms, lugworms and collembola. Future experiments should focus on more ecologically relevant taxa with emphasis on uptake, accumulation and key adverse effects like on survival rate, motility, growth and fertility as well as on the stability of the intestinal microbiome. Further studies with more than one test organism are important to foster our understanding of MP within certain food chains. Also long-term experiments might reveal adverse effects, which evolve slowly within populations. This may enable the assessement of the distribution and effects of MP within the food web and the resulting long-term impact on soil ecosystems.





## 5 Conclusion


Our review of 77 available studies on the impact of microplastic on the soil fauna shows an
alarming diversity and distribution of adverse effects within the soil tree of life. However, these
effects have to be considered carefully, as many experiments did not work with plastic
matching properties within natural soils and found adverse effects only at concentrations like
in highly contaminated soils or above. To elucidate effective concentrations for short and long-
term effects on soil faunal health, the most exact reproduction of plastic properties within the
soil matrix and natural living conditions of the test organisms is necessary. For future
experiments we therefore recommend to choose compositions of type, shape, size, grade of
weathering, leachability and coating with biofilms and other organic matter as expected in the
habitat to be examined. Furthermore, coming studies should include long-term exposure and
food chain experiments to get a better look at the effect of even smaller MP concentrations
and their enrichment within the food web. This may give us a better way of assessing the
impact of global microplastic contamination on e.g. soil biodiversity, soil carbon cycles and
soil quality.

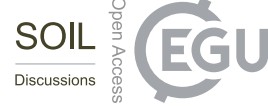

**Author contribution**

Frederick Büks developed the review concept, collected data and prepared the manuscript except for earthworms. Nicolette Loes van Schaik did all the work on earthworms. Martin Kaupenjohann supervised the whole study.

The authors declare that they have no conflict of interest.

**Acknowledgement**

Many thanks to Ivica Letunic, who kindly gave us access to phyloT and made our work in face of New Year much easier. I will not forget to invite you for a coffee.

**Competing interests**

The authors declare that they have no conflict of interest.





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
