# Peer review of "What do we know about how the terrestrial multicellular soil fauna reacts to microplastic?"

_SOIL, 2020_

## Referee Comment (RC1) · Anonymous Referee #1 · 16 Mar 2020

General comments:

In this manuscript, a systematic literature search was performed to review studies on interaction between edaphic organisms and plastic, from sub-$\mu$ to macro scale. This review is timely and of high relevance to the current discussion on plastic contamination in soils, its consequences for policy makers and for policy-informing research. The literature has been thoroughly and diligently screened and structured, so the reader gets a compact yet informative overview of the relevant metrics that help put the respective study into context. Also, the authors categorized the studies by taxa and grouped those according to phylogeny, providing a useful overview of the missing links of taxa not yet accounted for in research on uptake and adverse effects on soil organisms.

The manuscript is well written in a flow that makes even the somewhat dry summaries

of findings in the different taxonomic ranks easy to read and follow. The synthesis is well structured and clearly states the relevant points of findings, limitations, and research gaps.

This review merits publication in Soil after a few minor revisions I will be specifying below, since – to my knowledge - it is the first systematic review focusing on uptake, ingestion, and effects of (micro)plastics in the whole spectrum of soil fauna. This manuscript exhibits a good balance between depth and thematic distinction that is needed in a review worthwhile reading.

Specific comments:

Abstract

Lines 20-21: "Most of the studies applied MP concentrations similar to amounts in slightly to very heavily polluted soils." This sentence makes the reader expect that generally, the concentrations in the experimental environments are mostly the same as expected in the environment, but is this really the case? I would suggest showing the percentage of experiments with high microplastic exposure that is not representative of most soils.

Introduction

Line 53: Instead of "microbial decay", I'd suggest "processing by soil organisms", since this includes any process relevant for the generation of smaller plastic particles.

Line 61: I'd suggest changing the sentence to "intensive use of plastic mulching and sewage sludge", for the former, Huang et al. (2020) show an increase in microplastic by approx. 1 order of magnitude between fields with 5 and 24 continuous years of plastic mulching.

Line 95: Suggest changing "feed on" to "inadvertently ingest", otherwise it sounds like the organisms are actually able to metabolize the microplastics.

Search pattern

The cut-off dates (time period that was considered) of the search should be mentioned somewhere.

Figure 1: This figure shows the phylogenetic tree of edaphic fauna, rather than "edaphic tree of faunal life".

Data collection

Line 113-122: I've been having some difficulties understanding the search methodology and table 8 (table 8 should be moved at the appropriate place to become table 1). It would be great if the authors could re-word this, specifying:

- What does it mean that some combinations would have caused too much search effort?

- "Organism-plastic" is not a type-shape combination.

- What exactly does the number of studies in table 8 mean? The number of articles or single experiments (sometimes more than one taxon or plastic type is used in one article)?

Some articles are included that studied the uptake of macroplastics by organisms, mainly termites and ant species. It is reasonable to include these studies, but it should be mentioned more prominently, in the abstract and aims of the review, that macroplastics are included. Maybe also in the synthesis, a sentence about the proportions of experiments using macro-, micro-, and nanoplastic would be a helpful piece of information.

Tables 1-7: What does N/A mean in the tables? In some cases I assume "not analysed" (e.g., passive transport), but in other cases it should mean "not mentioned" (e.g., aging, coating, etc.) or "not observed" (e.g., measured adverse effects). I think this needs to be specified. Usually, N/A refers to "not applicable", but this doesn't fit in the tables.

Synthesis

Lines 549-550: Could you cite the studies that imitated weathering in the described way?

Lines 555-557: This is true, but it should be acknowledged that these additives are mainly present in commercial plastics, and therefore, mentioning of additives is not expected for "clean" microbeads specifically synthesized for the experiments. Nevertheless, the disadvantages of using these microbeads has been clearly discussed earlier in this section.

Conclusions

Line 620-621: I am a little concerned about describing the results as "alarming". Is it really? The following sentences actually refute this rather strong statement.

Lines 624-629: I would suggest changing the sentence to: "To elucidate[...], the most exact reproduction of plastic concentrations and properties [...]". However, the difficulty here is that very scarce data of limited quality is available on concentrations of microplastic in soils, so a range of concentrations need to be used for future experiments in order to match the "real world" concentrations in soil, while expecting a decrease in uncertainty in analytic results in the future. Especially in the lower size ranges (<100$\mu$m) quantification is currently challenging. Therefore, little is known about size distributions occurring in soils. It might be worth mentioning this dilemma in a sentence.

Technical corrections:

The following suggestions and corrections concern wording or spelling issues:

Line 79: replace "reasonable" with "likely"?

Line 84: replace "should be tested" with "has yet to be tested"?

Line 103: "This analysis is aimed to help assessing the influence of MP on the ecosystem services provided by diverse soil organisms."

Line 123: replace "appeared" with "was conducted"?

Line 131: replace "his purpose" with "their purpose" for gender mainstreaming

Line 144: replace "fine" with "smaller"

Line 148: replace "dependend" with "dependent"

Line 170: replace "soletary" with "solitary"

Line 238: replace "exhibit" with "exhibits"

Line 251: replace "peeled off" with "cast off"?

Line 385: replace "5 g kg-1" with "5000 mg kg-1" for consistency

Line 481: replace "measured for" with "observed in"?

Line 486: consider deleting "understandably", since the meaning of this word in this context is not clear to me.

Line 488: replace "checked" with "investigated"

Line 494: replace "concentration" with "concentrations"

Line 546: replace "remaines" with "remains"

Line 550: replace "therefor" with "therefore"

Line 660: replace "did not work with" with "did not use"

References

Huang, Y., Liu, Q., Jia, W., Yan, C. and Wang, J. (2020), "Agricultural plastic mulching as a source of microplastics in the terrestrial environment", Environmental pollution, Vol. 260, p. 114096.

---

## Referee Comment (RC2) · Anonymous Referee #2 · 7 Apr 2020

PD Dr. Werner Kratz, Free University of Berlin, Institute of Biology kratzw@zedat.fu-berlin.de

Microplastics are a new emerging soil pollutant of global importance. It is caused by fragmentation of larger plastic particles or by direct environmental emissions. Due to the widespread use and pathways of plastics, microplastics can be identified in the environment worldwide. The contamination of microplastic in terrestrial ecosystems is at least as big than in aquatic systems. Microplastics are small enough to ingest by a variety of soil organisms and plastic in nano-scale, they may cross biological barriers. In this review many studies were analyzed and evaluated regarding the effects of microplastics on soil and soil life. Based on the studies, significant changes in soil and soil properties were observed. The material properties (hydrophobicity, surfacevolume-ratio) of microplastics make it to a potential vector of chemical pollutants due to its sorption capacity with organic compounds. In addition, there are evidences for the adverse interaction of microplastics with terrestrial organisms and protists which fulfill important ecosystem services and functions. Microplastics can be transported by soil fauna (for example earthworms) and due to their small size, they also can reach the groundwater. Every change in soil or adverse effects on soil life affects the soil, which in turn is reflected in soil fertility. Microplastics per se are not a pollutant for the soil but its adverse effects could be a danger to the soil and thus to soil conservation. Despite the growing interest, previous studies are not sufficient to develop a better understanding of the problem of microplastics in soil, so further research is needed in the future. The authors of "What do we know about how the terrestrial multicellular soil fauna reacts to microplastic?" systematically reviewed studies on the transport, uptake and bioaccumulation of microplastic by soil organisms as well as adverse effects on their metabolism, microbiome, growth and survival parameters. The comprehensive review includes 77 studies until January 2020. Written in a compact and coherent manner it gives a low-threshold and widespread overview of the effects of microplastic on diverse branches of the edaphon, identifies gaps of former experimental setups and recommends key improvements. From my point of view, the manuscript corresponds to the interests of a large number of scientists, protagonists and stakeholders and fits very well into the growing discussion about the ecological effects of microplastics. For that reason, I strongly recommend the publication in SOIL.

In the following, I have only a few minor objections: Line 53: Is that only "microbial" decay? Line 507: "Preferably" instead of "preferrably". Lines 549-550: Is that proved that carboxylation of microspheres decreases hydrophobicity in an appreciable extent? Figure 1: The taxonomic group "further Panarthropods" is placed centrally, the other groups are not. Table 7: The last three experiments within this table were conducted by feeding the mice with a MP suspension. You might write "(food)" behind the concentration data as in the other tables. Table 8: Could you explain the meaning of the numbers within the table? Are these the numbers of experiments with the named type-shape

combinations?

---

## Author Comment (AC2) · 14 Apr 2020

Dear PD Dr. Werner Kratz. Thank you very much for your review. In the following I will try to answer your comments at my best. You will also find the corrections highlighted within our final response to the referees. Best regards, Frederick Büks

[1] Line 53: Is that only "microbial" decay? -> We agree, we will change this to "processing by soil organisms" as it is actually micro- as well as macroorganisms. [2] Figure 1: The taxonomic group "further Panarthropods" is placed centrally, the other groups are not. -> Done. [3] Table 7: The last three experiments within this table were conducted by feeding the mice with a MP suspension. You might write "(food)" behind the concentration data as in the other tables. -> Thanks a lot. Done. [4] Line 507:

[Figure]

"Preferably" instead of "preferrably". -> Done. [5] Table 8: Could you explain the meaning of the numbers within the table. Are these the numbers of experiments with the named type-shape combinations? -> Yes. Please see the answer to referee #1 (Table 8 is now Table 1). [6] Lines 549-507: Is that proved that carboxylation of microspheres decreases hydrophobicity in an appreciable extent? ->We ask the manufacturers of Polyciences Europe GmbH, a leading producer of PS microspheres, and they said no. We added this important information to the review.

———————————————

---

## Referee Comment (RC3) · Anonymous Referee #3 · 14 May 2020

Dear authors,

Investigations of the ecological effects of microplastics on organisms and ecosystems have long been limited to the limnic and marine realm, and many scientists (me included, I have to confess) underrated their importance for soils. This is changing now, and the number of actual papers on the subject is rapidly increasing. In my opinion, we are still in an early phase of this research, nevertheless, in your search you identified many published contributions on the subject. It thus seems to be a good point for a first review, and your contribution is both important, relevant, and timely. It is well placed in this journal an, once in adequate form, will find an interested readership.

I am sorry to say that I cannot support the publication of your paper in its present form. The main reason for this is that its presentation is not adequate for an international

audience - please find details below. But I encourage you to thoroughly work it through and present a much-improved version.

(1) First and foremost, please have the manuscript edited by a professional (!) native (!) biologist (!). The English of your text is largely understandable, but rough. Apart from annoying typos, I found sentences the meaning of which I only understood when trying to translate them to German (my native language). So, your text will heavily benefit from thorough native editing.

(2) Then, the text lacks conciseness, it is overly long. For example, I suggest to omit all biological/ecological details you provide when introducing a taxon. This is per se interesting, but not to the point here (except when the reader needs background to understand microplastic effects). Then, figure 1 does not contribute to the understanding of your presentation, omit it. And I do not think it necessary to present taxa for which there is no information available, especially if the taxa are of minor or no importance in soil (e.g. line 183ff, 205, 220, 227, 450ff) or if the literature is not on edaphic species (435ff). As a reviewer, you are of course required to address blind spots of research (thus pointing out important taxa that are missing in literature), but you need to better balance completeness with a concise presentation.

(3) I miss a convincing argumentation why you focus on multicellular animals (but then, you provide many details about bacteria, fungi, algae, plant roots in 72ff . . . omit this). A good line of reasoning could be that you follow up on the Rillig and Bonkowski (2018) paper.

(4) Please provide details of your literature search (123ff). When did you search? Which time span did you cover? Which search strings? Please consider the literature on meta-analyses how to properly specify these technical aspects.

(5) The first section of the paper needs much improvement, but the final parts (4.1, 4.2, 4.3, table 8) bring up very important aspects, great!

(6) line 636f: Please reconsider including your supervisor as a co-author. What "supervision" means is nowhere clearly defined, however, co-authorship is only justified for significant contributions to the manuscript. Honorary authorship violates the principle of scientific honesty.

---

## Referee Comment (RC4) · Maha DEEB (Referee) · 20 May 2020

Dear authors,

I would like to thank you for this work. I read several times and see no improvement that I could suggest. It's educational, really well presented, nicely written, and content is solid. I really appreciate how it opens new horizons for research in this field by highlighting the limits of the studies it mentions and summarizing thoroughly.

Based on this, I obviously recommend this work for publication.

---

## Editor Comment (EC1) · Fuensanta García-Orenes (Editor) · 21 May 2020

After read the manuscript and the interactive discussion, my recommendation for this work is to be accepted is SOIL as a new publication.

---

## Author Comment (AC4) · 25 May 2020

Dear Dr. Maha Deeb (referee #4) Thank you very much for the repeated check and your friendly report. Best regards, Dr. Frederick Büks
* * *

---

## Author Response (AR1)

**What do we know about how the terrestrial multicellular soil fauna reacts to microplastic?**

Frederick Büks[1], Nicolette Loes van Schaik[2], Martin Kaupenjohann[1]

[1]Chair of Soil Science, Dept. of Ecology, Technische Universität Berlin, 10587 Berlin, Germany
[2]Chair of Ecohydrology & Landscape Evaluation, Dept. of Ecology, Technische Universität Berlin, 10587 Berlin, Germany

*Correspondence to:* Frederick Büks (frederick.bueks@tu-berlin.de)

**Final responses to all referees plus marked-up version**

*Dear Referee #1*

*First I would like to express my sincere thanks to you for thoroughly reviewing our manuscript and for your very helpful and precise suggestions. In the following I will answer your points. Our corrections are marked-up with yellow numbers within the corrected manuscript at the end of this document.*

*Best regards,*
*Frederick Büks*

**Abstract**

**[1]** **Lines 20-21:** *"Most of the studies applied MP concentrations similar to amounts in slightly to very heavily polluted soils." This sentence makes the reader expect that generally, the concentrations in the experimental environments are mostly the same as expected in the environment, but is this really the case? I would suggest showing the percentage of experiments with high microplastic exposure that is not representative of most soils.*
Thanks a lot for this point. We now write: "About 58 % of the studies thereby use inappropriate concentrations or units, but 42 % applied MP concentrations similar to amounts in slightly to very heavily polluted soils."

**Introduction**

**[2]** **Line 53:** Instead of "microbial decay", I'd suggest "processing by soil organisms", since this includes any process relevant for the generation of smaller plastic particles.
Done.

**[3]** **Line 61:** I'd suggest changing the sentence to "intensive use of plastic mulching and sewage sludge", for the former, Huang et al. (2020) show an increase in microplastic by approx. 1 order of magnitude between fields with 5 and 24 continuous years of plastic mulching.
Done.

**[4]** **Line 95:** Suggest changing "feed on" to "inadvertently ingest", otherwise it sounds like the organisms are actually able to metabolize the microplastics.
Done and reference added.

**Search pattern**

**[5]** The cut-off dates (time period that was considered) of the search should be mentioned somewhere.
Information added to this chapter (see the answer to referee #3)

**[6]** Figure 1: This figure shows the phylogenetic tree of edaphic fauna, rather than "edaphic tree of faunal life".
Thank you. And done.

**Data collection**

**[7]** **Line 113-122:** I've been having some difficulties understanding the search methodology and table 8 (table 8 should be moved at the appropriate place to become table 1).
We moved the table to line 124 and mentioned that it contains the number of found studies. All table numbers were adjusted within the text.

It would be great if the authors could re-word this, specifying:

What does it mean that some combinations would have caused too much search effort?
It means e.g. that searching for a taxon only in combination with "PET" gives results for PET bottles for cultivation and experiments and also the "use" as pets, if the search is not case sensitive. We now tried to clarify this in our text.

"Organism-plastic" is not a type-shape combination.
Oh, yes, that's right. Corrected.

What exactly does the number of studies in table 8 mean? The number of articles or single experiments (sometimes more than one taxon or plastic type is used in one article)?
The number counts for how often type-shape combinations were used in all reviewed experimental setups independently of organism.

Some articles are included that studied the uptake of macroplastics by organisms, mainly termites and ant species. It is reasonable to include these studies, but it should be mentioned more prominently, in the abstract and aims of the review, that macroplastics are included.
Where macroplastics were used in the reviewed studies, the size was explicitly mentioned in the article text, so we do not see a necessity for elaborating the text. We did add a mention of macroplastics to the abstract.

Maybe also in the synthesis, a sentence about the proportions of experiments using macro-, micro-, and nanoplastic would be a helpful piece of information.
Now mentioned in "4.2 Limitations of previous studies"

**[8]** **Tables 1-7:** What does N/A mean in the tables? In some cases I assume "not analysed" (e.g., passive transport), but in other cases it should mean "not mentioned" (e.g., aging, coating, etc.) or "not observed" (e.g., measured adverse effects). I think this needs to be specified. Usually, N/A refers to "not applicable", but this doesn't fit in the tables.
In this work it means "(data) not available". We marked it at the tables.

**Synthesis**

**Lines 549-550:** Could you cite the studies that imitated weathering in the described way?
We did so. Tsunoda et al. (2010) artificially aged their plastic by soaking in hot water at 90°C for 21 days, and then it was sanded/scratched with medium-grade paper prior to the test. Gebhard and Forster (2018) incubated particles in seawater for 4 weeks to stimulate the formation of biofilms.

**[9]** **Lines 555-557:** This is true, but it should be acknowledged that these additives are mainly present in commercial plastics, and therefore, mentioning of additives is not expected for "clean" microbeads specifically synthesized for the experiments. Nevertheless, the disadvantages of using these microbeads has been clearly discussed earlier in this section.
Done.

**Conclusions**

**[10]** **Line 620-621:** I am a little concerned about describing the results as "alarming". Is it really? The following sentences actually refute this rather strong statement.
Replaced with "considerable".

**[11]** **Lines 624-629:** I would suggest changing the sentence to: "To elucidate [...], the most exact reproduction of plastic concentrations and properties [. . .]". However, the difficulty here is that very scarce data of limited quality is available on concentrations of microplastic in soils, so a range of concentrations need to be used for future experiments in order to match the "real world" concentrations in soil, while expecting a decrease in uncertainty in analytic results in the future. Especially in the lower size ranges (<100µm) quantification is currently challenging. Therefore, little is known about size distributions occurring in soils. It might be worth mentioning this dilemma in a sentence.
Done.

**[12]** Technical corrections:
All done.

Dear PD Dr. Werner Kratz  (referee #2)

*Thank you very much for your review. In the following I will try to answer your comments at my best. Our corrections are marked-up with green numbers within the corrected manuscript at the end of this document.*

*Best regards,*
*Frederick Büks*

**[1] Line 53:** Is that only "microbial" decay?
We agree, we will change this to "processing by soil organisms" as it is actually micro- as well as macroorganisms.

**[2] Figure 1:** The taxonomic group "further Panarthropods" is placed centrally, the other groups are not.
Done.

**[3] Table 7:** The last three experiments within this table were conducted by feeding the mice with a MP suspension. You might write "(food)" behind the concentration data as in the other tables.
Thanks a lot. Done.

**[4] Line 507:** "Preferably" instead of "preferrably".
Done.

**[5] Table 8:** Could you explain the meaning of the numbers within the table. Are these the numbers of experiments with the named type-shape combinations?
Yes. Please see the answer to referee #1 (Table 8 is now Table 1).

**[6] Lines 549-507:** Is that proved that carboxylation of microspheres decreases hydrophobicity in an appreciable extent?
We ask the manufacturers of Polyciences Europe GmbH, a leading producer of PS microspheres, and they said no. We added this important information to the review.

*Dear Referee #3*

*Thank you very much for your critical review of our manuscript. It has helped us to see some points which still need clarification. In the following, we want to explain how we propose to adjust our article based on the reviewer's comments and also explain why in some cases we do not agree with the reviewer's proposed changes. Our corrections are marked-up with purple numbers within the corrected manuscript at the end of this document.*

(1) First and foremost, please have the manuscript edited by a professional (!) native (!)biologist (!). The English of your text is largely understandable, but rough. Apart from annoying typos, I found sentences the meaning of which I only understood when trying to translate them to German (my native language). So, your text will heavily benefit from thorough native editing.

Rereading our article we did indeed see that some typos had escaped our notice. We are slightly surprised by the request of the reviewer to have the manuscript edited by a *"professional (!) native (!) biologist (!)"*. We rephrased some stiff sentences and corrected grammatical errors. If a proofreading is indeed wished, we will have a scientific translator (English native speaker) correct the article.

(2) Then, the text lacks conciseness, it is overly long. For example, I suggest to omit all biological/ecological details you provide when introducing a taxon. This is per se interesting, but not to the point here (except when the reader needs background to understand microplastic effects). Then, figure 1 does not contribute to the understanding of your presentation, omit it. And I do not think it necessary to present taxa for which there is no information available, especially if the taxa are of minor or no importance in soil (e.g. line 183ff, 205, 220, 227, 450ff) or if the literature is not on edaphic species (435ff). As a reviewer, you are of course required to address blind spots of research(thus pointing out important taxa that are missing in literature), but you need to better balance completeness with a concise presentation.

Your suggestion to omit the ecological presentation of some key taxa is understandable. If we would expect all readers to be well acquainted with the soil fauna, we would definitely go along with this. However, SOIL is a multi-disciplinary journal connecting a broad spectrum of soil scientists. Therefore, we think it is helpful to provide a short overview of information on the soil fauna, such as ecological functionalities (marker function, transport, degradation, habitat and food selection), which might influence how they cope with microplastics. We have critically gone through the article and here we summarize which parts we will shorten.

- [1] Proposal: We shortened the introduction of the springtail section, as it is indeed oversized.

For the same reason we illustrated the phylogenetic tree of soil life.

- Proposal: We would agree with moving it to the supplements in order to save space, in case this is wished.

We also do not fully agree with your suggestion to delete taxonomic groups that have not yet been subject of studies on microplastics. The reason is, that the aim of this work is not only to review effects on studied taxa, but also to show gaps of knowledge especially apart from the common model organisms. In fact, their importance for the current ecological research should be shortly mentioned.

- [2] Proposal: Unstudied taxa are still presented, but their importance for future research is now additionally mentioned in section 4.3 to better "balance completeness".
- [3] Proposal: We shortened the chapter about Onychophora.

*Potampoyrgus antipodarum* in fact is a benthic snail.

- [4] Proposal: We use this benthic species to show more clearly how inconsistent the few results for benthic and terrestrial snails are.

(3) I miss a convincing argumentation why you focus on multicellular animals (but then, you provide many details about bacteria, fungi, algae, plant roots in 72ff...omit this). A good line of reasoning could be that you follow up on the Rillig and Bonkowski (2018) paper.

The aim of this review is to depict the influence of microplastic contamination in soils to the soil fauna. But, to present a holistic view on the food web, we refer to microorganisms, plant roots and biofilms within the introduction section. Being large fields of knowledge on their own, these organisms are not part of the focus in this review, however they are food sources for meso- and macroorganisms and, thus, worthy of mention. Given that we only use 22 lines to describe these other parts of the phylogenetic tree of soil life, we think this is merited and wish to leave this part in the review.

Unfortunately, we do not understand how Rillig and Bonkowski (2018), a paper on soil protozoa, matches your point. We have read this paper and do mention it elsewhere in the review.

(4) Please provide details of your literature search (123ff). When did you search? Which time span did you cover? Which search strings? Please consider the literature on meta-analyses how to properly specify these technical aspects.

The search was applied between June 2019 and January 2020, repeated in the first week of January 2020 and covers publications until January 2020. The search strings result from combinations of taxon, plastic type and particle shape shown in Table 1 (formerly Table 8).

- **[5]** Proposal: Information added to section 2.

(5) Thank you very much for the positive note.

(6) line 636f: Please reconsider including your supervisor as a co-author. What "supervision" means is nowhere clearly defined, however, co-authorship is only justified for significant contributions to the manuscript. Honorary authorship violates the principle of scientific honesty.

We understand this point completely and agree that it is not good practice to include scientists who have not contributed significantly to a paper. We also acknowledge that supervision is a very broad term and would like to specify the contribution of Martin Kaupenjohann to the paper. **[6]** Martin Kaupenjohann was involved in the development of the idea and concept for this paper. During the literature reading and writing phase he has supported the work with frequent discussions of the contents of the article. And finally he has critically revised the manuscript.

Best regards,

Dr. Frederick Büks
Dr. Loes van Schaik
Prof. Dr. Martin Kaupenjohann

*Dear Dr. Maha Deeb (referee #4)*

*Thank you very much for the repeated check and your friendly report.*

*Best regards,*

*Dr. Frederick Büks*

[revised manuscript text omitted]

**Final responses to all referees plus marked-up version**

*Dear Referee #1*

*First I would like to express my sincere thanks to you for thoroughly reviewing our manuscript and for your very helpful and precise suggestions. In the following I will answer your points. Our corrections are marked-up with* yellow numbers *within the corrected manuscript at the end of this document.*

*Best regards,*
*Frederick Büks*

**Abstract**

**[1] Lines 20-21:** *"Most of the studies applied MP concentrations similar to amounts in slightly to very heavily polluted soils." This sentence makes the reader expect that generally, the concentrations in the experimental environments are mostly the same as expected in the environment, but is this really the case? I would suggest showing the percentage of experiments with high microplastic exposure that is not representative of most soils.*
Thanks a lot for this point. We now write: "About 58 % of the studies thereby use inappropriate concentrations or units, but 42 % applied MP concentrations similar to amounts in slightly to very heavily polluted soils."

**Introduction**

**[2] Line 53:** Instead of "microbial decay", I'd suggest "processing by soil organisms", since this includes any process relevant for the generation of smaller plastic particles.
Done.

**[3] Line 61:** I'd suggest changing the sentence to "intensive use of plastic mulching and sewage sludge", for the former, Huang et al. (2020) show an increase in microplastic by approx. 1 order of magnitude between fields with 5 and 24 continuous years of plastic mulching.
Done.

**[4] Line 95:** Suggest changing "feed on" to "inadvertently ingest", otherwise it sounds like the organisms are actually able to metabolize the microplastics.
Done and reference added.

**Search pattern**

**[5]** The cut-off dates (time period that was considered) of the search should be mentioned somewhere.
Information added to this chapter (see the answer to referee #3)

**[6]** Figure 1: This figure shows the phylogenetic tree of edaphic fauna, rather than "edaphic tree of faunal life".
Thank you. And done.

**Data collection**

**[7] Line 113-122:** I've been having some difficulties understanding the search methodology and table 8 (table 8 should be moved at the appropriate place to become table 1).
We moved the table to line 124 and mentioned that it contains the number of found studies. All table numbers were adjusted within the text.

It would be great if the authors could re-word this, specifying:

What does it mean that some combinations would have caused too much search effort?
It means e.g. that searching for a taxon only in combination with "PET" gives results for PET bottles for cultivation and experiments and also the "use" as pets, if the search is not case sensitive. We now tried to clarify this in our text.

"Organism-plastic" is not a type-shape combination.

Oh, yes, that's right. Corrected.

What exactly does the number of studies in table 8 mean? The number of articles or single experiments (sometimes more than one taxon or plastic type is used in one article)?

The number counts for how often type-shape combinations were used in all reviewed experimental setups independently of organism.

Some articles are included that studied the uptake of macroplastics by organisms, mainly termites and ant species. It is reasonable to include these studies, but it should be mentioned more prominently, in the abstract and aims of the review, that macroplastics are included.

Where macroplastics were used in the reviewed studies, the size was explicitly mentioned in the article text, so we do not see a necessity for elaborating the text. We did add a mention of macroplastics to the abstract.

Maybe also in the synthesis, a sentence about the proportions of experiments using macro-, micro-, and nanoplastic would be a helpful piece of information.

Now mentioned in "4.2 Limitations of previous studies"

**[8] Tables 1-7:** What does N/A mean in the tables? In some cases I assume "not analysed" (e.g., passive transport), but in other cases it should mean "not mentioned" (e.g., aging, coating, etc.) or "not observed" (e.g., measured adverse effects). I think this needs to be specified. Usually, N/A refers to "not applicable", but this doesn't fit in the tables.

In this work it means "(data) not available". We marked it at the tables.

**Synthesis**

**Lines 549-550:** Could you cite the studies that imitated weathering in the described way?

We did so. Tsunoda et al. (2010) artificially aged their plastic by soaking in hot water at 90°C for 21 days, and then it was sanded/scratched with medium-grade paper prior to the test. Gebhard and Forster (2018) incubated particles in seawater for 4 weeks to stimulate the formation of biofilms.

**[9] Lines 555-557:** This is true, but it should be acknowledged that these additives are mainly present in commercial plastics, and therefore, mentioning of additives is not expected for "clean" microbeads specifically synthesized for the experiments. Nevertheless, the disadvantages of using these microbeads has been clearly discussed earlier in this section.

Done.

**Conclusions**

**[10] Line 620-621:** I am a little concerned about describing the results as "alarming". Is it really? The following sentences actually refute this rather strong statement.

Replaced with "considerable".

**[11] Lines 624-629:** I would suggest changing the sentence to: "To elucidate [...], the most exact reproduction of plastic concentrations and properties [. . .]". However, the difficulty here is that very scarce data of limited quality is available on concentrations of microplastic in soils, so a range of concentrations need to be used for future experiments in order to match the "real world" concentrations in soil, while expecting a decrease in uncertainty in analytic results in the future. Especially in the lower size ranges (<100µm) quantification is currently challenging. Therefore, little is known about size distributions occurring in soils. It might be worth mentioning this dilemma in a sentence.

Done.

**[12]** Technical corrections:

All done.

Dear PD Dr. Werner Kratz  (referee #2)

*Thank you very much for your review. In the following I will try to answer your comments at my best. Our corrections are marked-up with green numbers within the corrected manuscript at the end of this document.*

*Best regards,*
*Frederick Büks*

**[1]** **Line 53:** Is that only "microbial" decay?
We agree, we will change this to "processing by soil organisms" as it is actually micro- as well as macroorganisms.

**[2]** **Figure 1:** The taxonomic group "further Panarthropods" is placed centrally, the other groups are not.
Done.

**[3]** **Table 7:** The last three experiments within this table were conducted by feeding the mice with a MP suspension. You might write "(food)" behind the concentration data as in the other tables.
Thanks a lot. Done.

**[4]** **Line 507:** "Preferably" instead of "preferrably".
Done.

**[5]** **Table 8:** Could you explain the meaning of the numbers within the table. Are these the numbers of experiments with the named type-shape combinations?
Yes. Please see the answer to referee #1 (Table 8 is now Table 1).

**[6]** **Lines 549-507:** Is that proved that carboxylation of microspheres decreases hydrophobicity in an appreciable extent?
We ask the manufacturers of Polyciences Europe GmbH, a leading producer of PS microspheres, and they said no. We added this important information to the review.

*Dear Referee #3*

*Thank you very much for your critical review of our manuscript. It has helped us to see some points which still need clarification. In the following, we want to explain how we propose to adjust our article based on the reviewer's comments and also explain why in some cases we do not agree with the reviewer's proposed changes. Our corrections are marked-up with* purple numbers *within the corrected manuscript at the end of this document.*

(1) First and foremost, please have the manuscript edited by a professional (!) native (!)biologist (!). The English of your text is largely understandable, but rough. Apart from annoying typos, I found sentences the meaning of which I only understood when trying to translate them to German (my native language). So, your text will heavily benefit from thorough native editing.
Rereading our article we did indeed see that some typos had escaped our notice. We are slightly surprised by the request of the reviewer to have the manuscript edited by a *"professional (!) native (!) biologist (!)"*. We rephrased some stiff sentences and corrected grammatical errors. If a proofreading is indeed wished, we will have a scientific translator (English native speaker) correct the article.

(2) Then, the text lacks conciseness, it is overly long. For example, I suggest to omit all biological/ecological details you provide when introducing a taxon. This is per se interesting, but not to the point here (except when the reader needs background to understand microplastic effects). Then, figure 1 does not contribute to the understanding of your presentation, omit it. And I do not think it necessary to present taxa for which there is no information available, especially if the taxa are of minor or no importance in soil (e.g. line 183ff, 205, 220, 227, 450ff) or if the literature is not on edaphic species (435ff). As a reviewer, you are of course required to address blind spots of research(thus pointing out important taxa that are missing in literature), but you need to better balance completeness with a concise presentation.
Your suggestion to omit the ecological presentation of some key taxa is understandable. If we would expect all readers to be well acquainted with the soil fauna, we would definitely go along with this. However, SOIL is a multi-disciplinary journal connecting a broad spectrum of soil scientists. Therefore, we think it is helpful to provide a short overview of information on the soil fauna, such as ecological functionalities (marker function, transport, degradation, habitat and food selection), which might influence how they cope with microplastics. We have critically gone through the article and here we summarize which parts we will shorten.
- [1] Proposal: We shortened the introduction of the springtail section, as it is indeed oversized.
For the same reason we illustrated the phylogenetic tree of soil life.
- Proposal: We would agree with moving it to the supplements in order to save space, in case this is wished.
We also do not fully agree with your suggestion to delete taxonomic groups that have not yet been subject of studies on microplastics. The reason is, that the aim of this work is not only to review effects on studied taxa, but also to show gaps of knowledge especially apart from the common model organisms. In fact, their importance for the current ecological research should be shortly mentioned.
- [2] Proposal: Unstudied taxa are still presented, but their importance for future research is now additionally mentioned in section 4.3 to better "balance completeness".
- [3] Proposal: We shortened the chapter about Onychophora.
*Potampoyrgus antipodarum* in fact is a benthic snail.
- [4] Proposal: We use this benthic species to show more clearly how inconsistent the few results for benthic and terrestrial snails are.

(3) I miss a convincing argumentation why you focus on multicellular animals (but then, you provide many details about bacteria, fungi, algae, plant roots in 72ff...omit this). A good line of reasoning could be that you follow up on the Rillig and Bonkowski (2018) paper.
The aim of this review is to depict the influence of microplastic contamination in soils to the soil fauna. But, to present a holistic view on the food web, we refer to microorganisms, plant roots and biofilms within the introduction section. Being large fields of knowledge on their own, these organisms are not part of the focus in this review, however they are food sources for meso- and macroorganisms and, thus, worthy of mention. Given that we only use 22 lines to describe these other parts of the phylogenetic tree of soil life, we think this is merited and wish to leave this part in the review.
Unfortunately, we do not understand how Rillig and Bonkowski (2018), a paper on soil protozoa, matches your point. We have read this paper and do mention it elsewhere in the review.

(4) Please provide details of your literature search (123ff). When did you search? Which time span did you cover? Which search strings? Please consider the literature on meta-analyses how to properly specify these technical aspects.

The search was applied between June 2019 and January 2020, repeated in the first week of January 2020 and covers publications until January 2020. The search strings result from combinations of taxon,  plastic type and particle shape shown in Table 1 (formerly Table 8).

- **[5]** Proposal: Information added to section 2.

(5) Thank you very much for the positive note.

(6) line 636f: Please reconsider including your supervisor as a co-author. What "supervision" means is nowhere clearly defined, however, co-authorship is only justified for significant contributions to the manuscript. Honorary authorship violates the principle of scientific honesty.

We understand this point completely and agree that it is not good practice to include scientists who have not contributed significantly to a paper. We also acknowledge that supervision is a very broad term and would like to specify the contribution of Martin Kaupenjohann to the paper. **[6]** Martin Kaupenjohann was involved in the development of the idea and concept for this paper. During the literature reading and writing phase he has supported the work with frequent discussions of the contents of the article. And finally he has critically revised the manuscript.

Best regards,

Dr. Frederick Büks
Dr. Loes van Schaik
Prof. Dr. Martin Kaupenjohann

*Dear Dr. Maha Deeb (referee #4)*

*Thank you very much for the repeated check and your friendly report.*

*Best regards,*

*Dr. Frederick Büks*

[revised manuscript text omitted]